# CROSS THE GAP: EXPOSING THE INTRA-MODAL MISALIGNMENT IN CLIP VIA MODALITY INVERSION

**Marco Mistretta**[1,*]**, Alberto Baldrati**[1,2,*]**, Lorenzo Agnolucci**[1,*]
**Marco Bertini**[1]**, Andrew D. Bagdanov**[1]
[1]University of Florence, Media Integration and Communication Center (MICC), Italy
[2]University of Pisa, Italy
`{name.surname}@unifi.it`

## ABSTRACT

Pre-trained multi-modal Vision-Language Models like CLIP are widely used off-the-shelf for a variety of applications. In this paper, we show that the common practice of individually exploiting the text or image encoders of these powerful multi-modal models is highly suboptimal for intra-modal tasks like image-to-image retrieval. We argue that this is inherently due to the CLIP-style inter-modal contrastive loss that does not enforce any intra-modal constraints, leading to what we call *intra-modal misalignment*. To demonstrate this, we leverage two optimization-based modality inversion techniques that map representations from their input modality to the complementary one without any need for auxiliary data or additional trained adapters. We empirically show that, in the intra-modal tasks of image-to-image and text-to-text retrieval, approaching these tasks *inter-modally* significantly improves performance with respect to intra-modal baselines on more than fifteen datasets. Additionally, we demonstrate that approaching a native inter-modal task (e.g. zero-shot image classification) *intra-modally* decreases performance, further validating our findings. Finally, we show that incorporating an intra-modal term in the pre-training objective or narrowing the *modality gap* between the text and image feature embedding spaces helps reduce the *intra-modal misalignment*. The code is publicly available at: `https://github.com/miccunifi/Cross-the-Gap`.

## 1 INTRODUCTION

In recent years the availability of massive, pre-trained Vision-Language Models (VLMs) has enabled a wide variety of applications ranging from zero-shot image segmentation (Zhou et al., 2022a; Lüddecke & Ecker, 2022) to visual question answering (Song et al., 2022; Parelli et al., 2023). These models are typically composed of independent image and text encoders which are simultaneously trained on massive corpora of image-text pairs to align the text and image embeddings of associated inputs. For example, the Contrastive Language-Image Pre-training (CLIP) model is trained on a corpus of 400M image-text pairs to map inputs from both modalities into a shared embedding space (Radford et al., 2021). CLIP is trained with an inter-modal contrastive loss that aims to maximize the similarity of corresponding image-text samples while minimizing the similarity with all the other examples within a batch.

Despite CLIP's shared embedding space, visual and textual features lie in distinct regions. This phenomenon, known as the *modality gap* (Liang et al., 2022), originates from model initialization, and the inter-modal contrastive loss preserves and worsens it during training. Moreover, we note that CLIP's contrastive training strategy focuses on *inter-modal* (*i.e.* image-text) similarities between paired samples and disregards *intra-modal* (*i.e.* image-image and text-text) similarities. Consequently, the intra-image and intra-text similarities between CLIP representations might not faithfully correspond to those of the actual images or texts, as depicted in the left section of Fig. 1. We refer to this issue as *intra-modal misalignment*. A simple experiment aimed at quantifying this problem is presented in Appendix B.

---

*These authors contributed equally to this work.

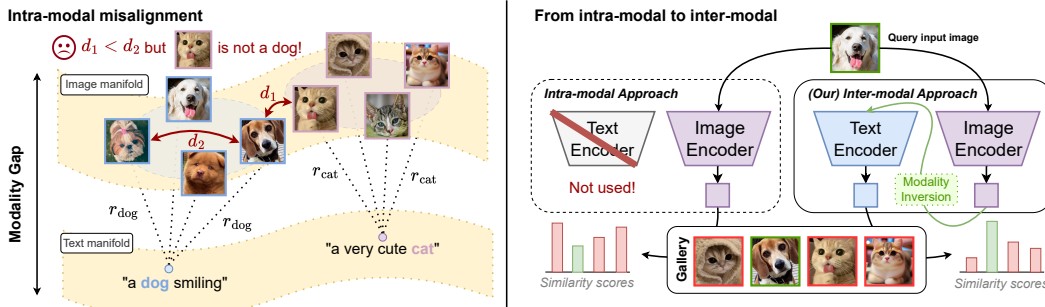

Figure 1: Motivation and overview. **Left:** The *inter-modal* contrastive loss used in pretraining enforces paired images and texts to be at a given distance $r$ (*i.e.* $r_{dog}$ and $r_{cat}$) but does not encourage *intra-modal* alignment. Consequently, intra-modal similarity scores might not correspond to those of actual images and texts (*i.e.* $d_1 < d_2$). **Right:** We show that the common practice of individually exploiting only one encoder is suboptimal and approaching intra-modal tasks (*e.g.* image-to-image retrieval) *inter-modally* via modality inversion improves performance.

Aspects of this misalignment have been accounted for in the limited scope of zero- and few-shot image classification (Udandarao et al., 2023; Yi et al., 2024). However, many recent works overlook this phenomenon and still employ CLIP representations for intra-modal comparison, which leads to suboptimal similarity measurements. Examples range from KNN-based image classification (Geirhos et al., 2024) to text-to-image generation (Gal et al., 2022; Ruiz et al., 2023) and video synthesis (Esser et al., 2023; Zhang et al., 2024). For instance, Esser et al. (2023) measure the temporal consistency of generated videos via intra-modal CLIP similarity between consecutive frames.

In this paper we argue that relying on intra-modal similarities computed using pre-trained CLIP encoders is inherently suboptimal. To support this we conduct an extensive study of the behavior of intra-modal similarities on the intra-modal tasks of image-to-image and text-to-text retrieval. We perform this analysis by transforming intra-modal tasks into inter-modal ones to leverage CLIP's inter-modal alignment.

Specifically, we map features from their *native* modality (*i.e.* the same as the input) into their *complementary* one. We refer to this process as *modality inversion*. To perform modality inversion we adapt Optimization-based Textual Inversion (OTI) (Baldrati et al., 2023) and introduce Optimization-based Visual Inversion (OVI). OTI and OVI are iterative modality inversion strategies that map image features into text features and vice versa while keeping the encoders frozen. These techniques operate at the single-feature level, *i.e.* they do not require external data nor the training of a mapping network.

Our experiments show that tackling intra-modal tasks inter-modally via modality inversion – as illustrated in the right side of Fig. 1 – outperforms intra-modal baselines on more than fifteen datasets. To additionally support our claim that this performance improvement stems from inter-modal alignment and not the modality inversion process itself, we transform inter-modal tasks into intra-modal ones. Specifically, we show that applying modality inversion to the inherently inter-modal zero-shot image classification task yields *worse* performance than the inter-modal baseline.

Moreover, we investigate whether the inclusion of an intra-modal loss during image-text contrastive pre-training reduces intra-modal misalignment. For this analysis we use SLIP (Mu et al., 2022), which adds just such an intra-modal loss to improve the alignment within the image embedding space. Results confirm that adding intra-modal loss terms during the pre-training of VLMs significantly mitigates intra-modal misalignment. Finally, we study the relation between the modality gap phenomenon and the intra-modal misalignment. In particular, similar to Liang et al. (2022) we fine-tune CLIP to reduce the modality gap and observe a decrease in the performance of approaching intra-modal tasks inter-modally. This indicates that a narrower modality gap diminishes the impact of intra-modal misalignment.

The main contributions of this work are:

- we conduct a thorough and comprehensive study of CLIP's intra-modal misalignment, and our analyses show that the common practice of relying on intra-modal similarities computed through pre-trained CLIP encoders is inherently suboptimal.
- we propose to transform intra-modal tasks to inter-modal ones via modality inversion to exploit CLIP's inter-modal alignment. To this end we introduce OVI, a single-feature level modality inversion strategy that maps textual features into the image embedding space;
- we conduct extensive experiments that show that approaching intra-modal tasks inter-modally significantly outperforms intra-modal baselines on more than fifteen datasets; and
- we demonstrate that adding intra-modal loss terms during VLM pre-training or reducing the modality gap mitigates the impact of intra-modal misalignment.

## 2 RELATED WORK

**Contrastively trained Vision-Language Models.** VLMs have become increasingly popular for their ability to learn aligned representations across visual and textual modalities (Radford et al., 2021; Jia et al., 2021; Zhai et al., 2022; 2023; Mu et al., 2022; Li et al., 2021). This alignment enables VLMs to be used in a broad variety of downstream tasks, including image-text retrieval and zero-shot image classification, by projecting images and text into a shared feature space.

The most prominent example is CLIP (Radford et al., 2021), which maximizes the similarity between paired images and text captions while minimizing the similarity with the other samples in the batch. SigLIP (Zhai et al., 2023), on the other hand, employs a sigmoid-based contrastive loss that considers only the single image-text pairs while neglecting the other samples in the same batch. More recently, several approaches have extended the CLIP-style contrastive loss by incorporating intra-modal similarities into the training objectives (Mu et al., 2022; Li et al., 2021). For instance, SLIP (Mu et al., 2022) integrates a self-supervised component that maximizes the similarity between different augmentations of the same image, with a strategy akin to SimCLR (Chen et al., 2020).

**The modality gap in multi-modal models.** Liang et al. (2022) demonstrated a consistent phenomenon affecting VLMs known as the *modality gap*. This refers to the separation between feature embeddings of different modalities (*e.g.* text and images) within their shared representation space (Liang et al., 2022). The modality gap arises due to both model initialization and the contrastive learning objective used during training. At initialization, independent encoders for each modality produce embeddings that are restricted to distinct regions (or cones) within the representation space. During training, the contrastive learning process preserves and worsens this separation. Several works have studied the causes and implications of the modality gap in CLIP (Shi et al., 2023; Schrodi et al., 2024; Zhang et al., 2023). Schrodi et al. (2024) analyzed the embedding space and demonstrated that a minimal number of embedding dimensions – often as few as two – are sufficient to perfectly separate the image and text modalities.

**Intra-modal misalignment.** Some studies have investigated the problem of misaligned intra-modal embedding distances within the context of zero- and few-shot image classification (Udandarao et al., 2023; Yi et al., 2024). To address this, Udandarao et al. (2023) propose mitigating the issue by computing similarities in the image-text space, rather than working exclusively with image embeddings, thereby leveraging the inter-modal nature of the feature representations. Similarly, CODER (Yi et al., 2024) introduces an enhanced image representation technique based on measuring distances between images and their neighboring texts within CLIP's embedding space.

**Our contribution with respect to the state-of-the-art.** While these prior works have addressed various aspects of intra-modal and inter-modal relationships within VLMs, their scope remains limited, often focusing on specific tasks, datasets, or narrow perspectives on the modality gap and its effects. None of these studies comprehensively investigate the fundamental nature of the intra-modal versus inter-modal similarities across diverse tasks and datasets, nor do they fully explore the potential performance improvements achievable by leveraging inter-modal comparisons for intra-modal problems. The motivation behind our work is to shed light on the phenomenon of intra-modal misalignment, and its relationship to the modality gap, and to demonstrate the importance of either ensuring intra-modal alignment during pre-training or comparing solely representations that belong to different modalities.

## 3 CLIP PRELIMINARIES

CLIP is a vision-language model trained to align images and textual captions in a shared embedding space (Radford et al., 2021). It consists of an image encoder $f_\theta$ and a text encoder $g_\phi$. Given an image $I$, the image encoder extracts its feature representation $f_\theta(I) \in \mathbb{R}^d$, where $d$ is the size of the shared embedding space. Likewise, for a given textual caption $Y$, first a word embedding layer $E_v$ maps each tokenized word to the token embedding space $\mathcal{V}$. Then, the text encoder $g_\phi$ generates the textual feature representation $g_\phi(E_v(Y)) \in \mathbb{R}^d$.

When using a Vision Transformer (ViT) (Dosovitskiy et al., 2020) as the visual encoder $f_\theta$, the encoding process begins by splitting the image into $U$ fixed-size non-overlapping patches. Each patch is then transformed into a corresponding patch embedding $\{w_1, w_2, \ldots, w_U\}$ through a linear projection by the patch embedding layer $E_w$, where each $w_i$ resides in the patch embedding space $\mathcal{W}$. A learnable class (CLS) token $c$ is concatenated with the patch embeddings, resulting in the input to the vision transformer being $\bar{I} = \{c, w_1, w_2, \ldots, w_U\}$. Finally, the CLS token of the final transformer layer is projected into the shared embedding space via a linear projection to obtain the final representation $f_\theta([c, E_w(I)]) = f_\theta(\bar{I}) \in \mathbb{R}^d$. For brevity, when unnecessary we will omit both the patch embedding layer $E_w$ and the token embedding layer $E_v$, and use the simplified notations $f_\theta(I)$ instead of $f_\theta([c, E_w(I)])$ and $g_\phi(Y)$ instead of $g_\phi(E_v(Y))$.

Given a batch of image-caption pairs $B = \{(I_n, Y_n)\}_{n=1}^N$, CLIP aims to maximize the cosine similarity for the $N$ correct pairs while minimizing it for the $N^2 - N$ other pairs. This is achieved by optimizing a symmetric, multi-class N-pair contrastive loss (Sohn, 2016). Let $\psi_I^n = f_\theta(I_n)$ and $\psi_T^n = g_\phi(E_v(Y_n))$ denote the image and text embeddings, respectively. The CLIP loss is:

$$\mathcal{L}_{\text{CLIP}} = -\frac{1}{N} \sum_{n=1}^N \left( \log \frac{\exp(c(\psi_I^n, \psi_T^n)/\tau)}{\sum_{m=1}^N \exp(c(\psi_I^n, \psi_T^m)/\tau)} + \log \frac{\exp(c(\psi_T^n, \psi_I^n)/\tau)}{\sum_{m=1}^N \exp(c(\psi_T^n, \psi_I^m)/\tau)} \right), \quad (1)$$

where $c(\cdot, \cdot)$ denotes the cosine similarity, and $\tau$ is a temperature parameter. As shown by Liang et al. (2022), Eq. (1) leads to a measurable separation between embeddings of the different modalities, creating what is known as the *modality gap*. This gap is significantly affected by the temperature $\tau$, with a larger gap occurring as the temperature decreases.

Note that CLIP's training loss focuses exclusively on inter-modal similarities between paired samples while neglecting intra-modal similarities. For example, consider an image feature anchor $\psi_I$ and two distinct text features $\psi_T^1$ and $\psi_T^2$ expressing the same concept. The loss enforces both $\psi_T^1$ and $\psi_T^2$ to be at a cosine distance $r$ from $\psi_I$, where the cosine distance is defined as $d(\psi_A, \psi_B) = 1 - c(\psi_A, \psi_B)$. This is equivalent to $d(\psi_I, \psi_T^1) = d(\psi_I, \psi_T^2) = r$, meaning the text embeddings lie on a hypersphere of radius $r$ centered at $\psi_I$. The absence of intra-modal constraints leaves the alignment between $\psi_T^1$ and $\psi_T^2$ remains uncalibrated; thus, we have $0 \leq d(\psi_T^1, \psi_T^2) \leq 2r$. This indicates that, while both text features are equidistant from the image feature, their intra-modal similarity is not constrained in any way, leading to intra-modal misalignment. We argue that such a misalignment must either be mitigated via additional intra-modal losses during pre-training or must be compensated by tackling intra-modal tasks inter-modally.

## 4 FROM INTRA-MODAL TO INTER-MODAL VIA MODALITY INVERSION

Due to the modality gap, images and text features lie in distinct regions in CLIP's shared embedding space. Previous work introduced modality inversion techniques to map features from the native modality to the complementary one (Ramesh et al., 2022; Patel et al., 2024; Li et al., 2023). For instance, Ramesh et al. (2022) trains a diffusion model to generate CLIP's image features from text captions for text-to-image generation.

Our goal is to demonstrate that tackling intra-modal tasks in an inter-modal way outperforms intra-modal baselines. To this end, we propose to employ a modality inversion strategy to derive representations that exploit both native and complementary modality encoders. However, existing modality inversion techniques rely on external data or the training of a mapping network, making the inversion process dependent on external factors (Ramesh et al., 2022; Patel et al., 2024; Li et al., 2023).

To minimize the impact of external biases in our analysis, we choose to rely on two modality inversion strategies that operate at a single-feature level, *i.e.* that map each individual feature to its complementary modality without the need for any external resources. Specifically, we adapt Optimization-based Textual Inversion (OTI) (Baldrati et al., 2023; Agnolucci et al., 2024) and we introduce Optimization-based Visual Inversion (OVI) to map an image to the text embedding space and vice versa while keeping the encoders frozen. Both are iterative and optimization-based approaches. The core concept behind OTI and OVI is to learn vectors of trainable parameters that are passed through the encoder of the *complementary* modality to yield features aligned with the representations of the *native* modality encoder. By optimizing these input vectors while keeping the encoder weights fixed, we ensure that the output features retain the pre-training alignment. In the following we define OTI and OVI for CLIP, but they can be applied to any VLM that maps images and texts into a shared embedding space. To make our analysis more comprehensive, in Appendix G we present two additional experiments in which modality inversion is performed using a pre-trained captioner or an adapter.

## 4.1 Optimization-based Textual Inversion (OTI)

Starting from an image $I$, OTI involves iteratively optimizing a set of $R$ pseudo-tokens $v^* = \{v_1^*, v_2^*, \ldots, v_R^*\}$, with $v_i^* \in \mathcal{V}$ for $i \in \{1, \ldots, R\}$, for a given number of optimization steps $S$. We refer to $v^*$ as pseudo-tokens since they belong to the token embedding space $\mathcal{V}$ but are not associated with any existing words. Algorithm 1 in Appendix A shows the pseudo-code of OTI.

The pseudo-tokens $v^*$ are randomly initialized and concatenated with the template sentence "a photo of" to form $\overline{Y}_{v^*} = [E_v(\text{"a photo of"}), v^*]$ input into the CLIP text encoder $g_\phi$ to obtain $\psi_T = g_\phi(\overline{Y}_{v^*})$. Then we extract the features of the image $I$ with the CLIP image encoder $f_\theta$, resulting in $\psi_I = f_\theta(I)$. Since we aim to obtain a textual feature representation $\psi_T$ that captures the informative content of $I$, we minimize the gap between image and text features via a cosine loss:

$$\mathcal{L}_{\cos} = 1 - c(\psi_I, \psi_T). \tag{2}$$

Note that while we adapt OTI from Baldrati et al. (2023) our goal is significantly different. Their work focuses on deriving a single pseudo-token that captures the informative content of the image $I$ and can interact with existing words to form meaningful sentences (e.g., "a photo of $v^*$ that is running …"), thus they use OTI for combining inputs from both modalities. In contrast, we use OTI purely as a mapping technique from visual to textual features. We do not focus on the pseudo-tokens themselves but aim to obtain a final representation that effectively captures the content of the image $I$. Additionally, the original OTI technique employs a regularization loss that exploits an auxiliary vocabulary to constrain the pseudo-token to reside in CLIP's token embedding space. However we are not interested in using the learned $v^*$ in different contexts – and more importantly, we aim to avoid influencing the inversion process with external data. For this reason we do not use a regularization loss.

## 4.2 Optimization-based Visual Inversion (OVI)

We propose the OVI approach to map text features from the CLIP text embedding space to the visual embedding space. Note that since OVI learns vectors of trainable parameters in the patch embedding space $\mathcal{W}$, it can be applied only to ViT-based image encoders.

Given a sentence $Y$, we first extract its text features $\psi_T = g_\phi(E_v(Y))$. OVI then optimizes a set of $P$ randomly initialized pseudo-patches $w^* = \{w_1^*, \ldots, w_P^*\}$, where each $w_i^* \in \mathcal{W}$. This optimization is performed for a fixed number of optimization steps $S$. Similarly to the terminology introduced in Sec. 4.1, we refer to $w^*$ as pseudo-patches since they belong to the patch embedding space $\mathcal{W}$ but are not associated with any existing image. Algorithm 2 in Appendix A illustrates the pseudo-code of the OVI method.

Since the ViT employs learned positional embeddings, the number of input patches $U$ to the image encoder is fixed. Consequently, when $P < U$ directly using $w^*$ as input is impossible. In such cases, we apply nearest-neighbor interpolation to repeat the pseudo-patches and match the required number of $U$ patches.

Specifically, given the pre-trained CLS token $c$, the input to the ViT is given by:

$$\bar{I}_{w^*} = \{c, \underbrace{w_1^*, w_1^*, \ldots, w_1^*}_{H_1 \text{ times}}, \underbrace{w_2^*, w_2^*, \ldots, w_2^*}_{H_2 \text{ times}}, \ldots, \underbrace{w_P^*, w_P^*, \ldots, w_P^*}_{H_P \text{ times}}\}, \tag{3}$$

where $H_1, \ldots, H_P$ represent the number of times each pseudo-patch is repeated, and $H_1 + \ldots + H_P = U$. The specific values are given by the nearest-neighbor interpolation. Finally, the input $\bar{I}_{w^*}$ is passed through CLIP's image encoder to obtain the features $\psi_I = f_\theta(\bar{I}_{w^*})$. To obtain a visual feature representation $\psi_I$ that captures the informative content of $Y$, we minimize the gap between the image and text features using the same cosine-based loss in Eq. (2).

### 4.3  CROSSING THE MODALITY GAP WITH OTI AND OVI

The goal of OTI and OVI is to map features from the native modality into corresponding features in the complementary modality. We observe that in cases where the loss $\mathcal{L}_{\cos}$ approaches zero, the complementary features converge to the native ones, thus drifting into the native modality embedding manifold. This undermines the goal of leveraging CLIP's image-text alignment.

For OTI, in our experiments the loss never approaches zero – within a reasonable number of optimization steps – when considering a single pseudo-token (*i.e.* $R = 1$). We argue that this stems from the strong inductive biases of the frozen encoders and the modality gap, making it challenging for a single pseudo-token to bridge the distance between image and text representations. Nevertheless, the OTI-inverted features retain the informative content of the corresponding image. As a result, the potential drift related to $\mathcal{L}_{\cos}$ does not pose a significant issue, and inter-modal alignment is preserved. In all experiments we use $R = 1$ unless stated otherwise.

Also for OVI we observe that the loss only approaches zero when the number of pseudo patches $P$ is relatively large. Unlike OTI, we find that for some experiments a single pseudo-patch (*i.e.* $P = 1$) is insufficient for embedding the informative content of the corresponding text. We believe that this discrepancy stems from the inherent differences between images and texts. Specifically, in textual inputs a single word (or pseudo-token) can significantly alter the meaning of a sentence. For instance, the sentences "a photo of a building" and "a photo of a dog" convey completely different meanings, despite differing by only one word. In contrast, a single (pseudo-)patch has less influence on the overall semantic content of an image. Therefore, while a single pseudo-token is enough for an effective modality inversion with OTI, more pseudo-patches may be necessary when applying OVI. Consequently, in our experiments, we employ a number of pseudo-patches $P$ ranging from 1 to 4, based on the considered model (see Appendix D for more details). For such values, the pseudo-patches effectively embed the informative content of the input text. Moreover, the inter-modal alignment is maintained and the drift does not constitute a significant problem.

## 5  EXPERIMENTAL RESULTS

Here we report on a broad range of experiments supporting our claims. We first evaluate two intra-modal tasks: image-to-image and text-to-text retrieval. We show that transforming intra-modal tasks into inter-modal ones via OTI and OVI consistently improves performance by better aligning with the original CLIP training objective. To confirm that this outcome does not stem from the modality inversion process itself, we evaluate a natively inter-modal task – zero-shot image classification – and show that making it intra-modal hinders the performance. Finally, we analyze the behavior of modality inversion techniques, and we study how adding intra-modal loss terms during VLM pre-training or narrowing the modality gap affects intra-modal misalignment. In the following, we denote as *inter-modal approaches* those involving inter-modal similarity comparisons, *i.e.* similarity comparisons between features of two different modalities (such as image-text, OTI-image, and OVI-text). Conversely, *intra-modal approaches* refer to methods that employ intra-modal similarity comparisons (such as image-image, text-text, OTI-OTI, and OVI-OVI).

To ensure a comprehensive analysis, we experiment using multiple CLIP models with different backbones and pre-training datasets. We also consider SigLIP to prove that our observations are not specific to the CLIP loss but generalize to other inter-modal contrastive losses. Specifically, we use OpenAI CLIP with ViT-B/32 and ViT-L/14 backbones, OpenCLIP (OPEN) pre-trained on the DataComp dataset (Gadre et al., 2024) with the same backbones, and SigLIP-B/16. Implementation details and description of all datasets used are given in Appendices A and F.

Table 1: Performance (mAP) evaluation on the image-to-image retrieval task. Blue rows indicate the usage of OTI-inverted features, while white rows refer to the intra-modal baseline. ✓ and ✗ denote inter-modal and intra-modal approaches, respectively.

| | Backbone | Inter modal | CUB | SOP | ROxford | RParis | Cars | Pets | Flowers | Aircraft | DTD | EuroSAT | Food101 | SUN397 | Caltech | UCF101 | ImageNet | Average |
|---|---|---|---|---|---|---|---|---|---|---|---|---|---|---|---|---|---|---|
| CLIP | B/32 | ✗ | 22.9 | 34.4 | 42.6 | 67.9 | 24.6 | 30.5 | 62.0 | **14.5** | 28.1 | **47.9** | 32.3 | 34.3 | 77.1 | 47.1 | 21.4 | 39.2 |
| | | ✓ | **24.6** | **35.1** | **43.0** | **70.3** | **28.0** | **37.5** | **62.6** | 14.4 | **31.9** | 47.2 | **34.7** | **36.3** | **79.9** | **48.6** | **23.8** | 41.2 |
| | L/14 | ✗ | 43.0 | 40.8 | 57.5 | 76.9 | 43.3 | 47.3 | 84.0 | 25.8 | 34.1 | **59.0** | 53.0 | 39.1 | 83.2 | 60.0 | 33.1 | 52.0 |
| | | ✓ | **47.1** | **41.2** | **62.4** | **77.1** | **50.5** | **56.0** | **86.0** | **27.1** | **37.7** | 56.3 | **55.9** | **43.5** | **87.3** | **62.8** | **38.2** | 55.3 |
| OPEN | B/32 | ✗ | 32.1 | 43.0 | 50.8 | 74.7 | 46.7 | 44.1 | 77.0 | 19.6 | 36.9 | **56.4** | 39.6 | 36.2 | 82.3 | 45.7 | 24.7 | 47.3 |
| | | ✓ | **34.3** | **44.5** | **54.4** | **75.8** | **50.5** | **50.5** | **78.0** | **20.1** | **40.9** | 54.5 | **42.9** | **37.8** | **83.3** | **48.2** | **27.3** | 49.5 |
| | L/14 | ✗ | 56.4 | 50.7 | 69.0 | 83.9 | 65.4 | 61.4 | 91.6 | 32.5 | 40.4 | **63.8** | 61.1 | 42.2 | 86.9 | 62.6 | 38.8 | 60.4 |
| | | ✓ | **58.9** | **51.9** | **73.2** | **87.7** | **72.6** | **67.3** | **92.7** | **34.3** | **44.3** | 63.1 | **65.2** | **45.8** | **89.7** | **64.7** | **42.6** | 63.6 |
| SigLIP | B/16 | ✗ | 39.4 | 49.9 | 50.6 | 73.9 | 65.7 | 56.5 | 87.0 | **37.9** | 39.9 | 52.4 | 56.3 | 42.8 | 87.3 | **56.7** | 35.9 | 55.5 |
| | | ✓ | **41.8** | **53.0** | **55.2** | **79.1** | **71.8** | **64.2** | **89.7** | 37.6 | **43.3** | **52.9** | **59.0** | **43.6** | **88.9** | 54.9 | **38.8** | 58.3 |

## 5.1 IMAGE-TO-IMAGE RETRIEVAL

Pre-trained CLIP image encoders are often used to extract features for image-to-image similarity comparisons. For this reason, we study the image-to-image retrieval task.

**Experiment design.** The objective is to retrieve images from a gallery that are visually similar to a given query image. We consider a total of 15 datasets commonly employed for image-to-image retrieval and image classification. We consider two strategies. In the first, which we call *intra-modal*, we directly compare the features of the query image with those of the gallery. In the second, we transform the intra-modal image-to-image retrieval task into an inter-modal one by applying OTI to the query image. Then, we use the resulting OTI-inverted features to query the gallery.

**Results.** We report the results in Tab. 1. Approaching the task *inter-modally* using the OTI-inverted features outperforms the intra-modal baseline, achieving an average absolute improvement ranging from 2% to 3%. Specifically, we observe that the performance gain is obtained across a large variety of datasets with different class granularity and diverse domains, spanning from birds (CUB) to monuments (RParis). Moreover, we notice that the intra-modal misalignment phenomenon is independent of the pre-training dataset (CLIP vs. OpenCLIP) and pre-training contrastive loss (CLIP/OpenCLIP vs. SigLIP) since the performance improvement is consistent across all the considered VLMs. Finally, we note that OTI-inverted features cannot contain more informative content than the native ones – used by the intra-modal baseline – since they are obtained simply by mapping them to the complementary modality at a single-feature level and without using any external resources. Thus, the observed improvement is solely attributable to inter-modal alignment rather than to a more enriched representation.

## 5.2 TEXT-TO-TEXT RETRIEVAL

Although text features from pre-trained CLIP models are not commonly used for text-to-text tasks, we believe that it is important to show that our findings also apply to text-to-text comparisons.

**Experiment design.** Using the CLIP text encoder for text-only tasks presents several challenges. Specifically, the CLIP text encoder is trained on short, descriptive texts. As a result, using it for tasks such as sentiment analysis or text classification, which involve longer texts and abstract concepts, results in a mismatch with the pre-training data. Moreover, VLMs have a limited input token capacity (*e.g.* 77 tokens for CLIP), which makes them unsuitable for longer texts. To avoid these problems, we formulate an intra-modal text-to-text retrieval task using image captioning datasets. Specifically, we select datasets in which each sample consists of an image and multiple associated captions (*e.g.* Flickr30K (Plummer et al., 2015)). These captions are comparable to those used in VLM training and are short enough to avoid token limit issues. We ignore the images and use the first caption associated with each image as the query text. The goal is to retrieve the other captions related to the same image from a gallery of all captions in the dataset. In Appendix G, we also report experimental results on purely textual text-to-text retrieval datasets. To address the issue of texts exceeding the input token limit of VLMs, we use an LLM to summarize the texts.

Table 2: **Left**: Performance (mAP) evaluation on the text-to-text retrieval task. Purple rows indicate the usage of OVI-inverted features, while white rows refer to the intra-modal baseline. **Right**: Performance (accuracy) evaluation on the zero-shot image classification task. Blue rows indicate the usage of OTI-inverted features, while white rows refer to the inter-modal baseline. ✓ and X denote inter-modal and intra-modal approaches, respectively.

| | Backbone | Inter modal | Flickr30k | COCO | nocaps | Average |
|---|---|---|---|---|---|---|
| CLIP | B/32 | X | 51.7 | 26.2 | 35.1 | 37.7 |
| | | ✓ | 54.8 | 28.3 | 38.8 | 40.6 |
| | L/14 | X | 52.3 | 26.7 | 35.7 | 38.2 |
| | | ✓ | 54.9 | 29.4 | 39.5 | 41.3 |
| OPEN | B/32 | X | 58.0 | 30.0 | 40.3 | 42.8 |
| | | ✓ | 60.2 | 32.0 | 43.6 | 45.3 |
| | L/14 | X | 61.0 | 31.8 | 42.5 | 45.1 |
| | | ✓ | 63.6 | 33.0 | 44.5 | 47.0 |
| Sig LIP | B/16 | X | 56.7 | 27.0 | 38.6 | 40.8 |
| | | ✓ | 60.1 | 29.6 | 43.4 | 44.4 |

| | Backbone | Inter modal | Cars | Pets | Flowers | Aircraft | DTD | EuroSAT | Food101 | SUN397 | Caltech | UCF101 | ImageNet | Average |
|---|---|---|---|---|---|---|---|---|---|---|---|---|---|---|
| CLIP | B/32 | ✓ | 60.4 | 87.5 | 67.0 | 19.1 | 43.6 | 45.2 | 80.5 | 62.0 | 91.2 | 62.0 | 62.1 | 61.9 |
| | | X | 54.5 | 80.9 | 61.2 | 17.3 | 41.8 | 39.4 | 75.3 | 54.6 | 83.7 | 58.5 | 53.6 | 56.4 |
| | L/14 | ✓ | 76.8 | 93.6 | 79.3 | 32.5 | 53.0 | 58.1 | 91.0 | 67.6 | 94.9 | 74.2 | 73.5 | 72.2 |
| | | X | 72.1 | 89.8 | 73.1 | 29.4 | 52.3 | 56.4 | 87.6 | 62.4 | 90.2 | 71.3 | 68.0 | 68.4 |
| OPEN | B/32 | ✓ | 88.4 | 90.3 | 73.5 | 24.4 | 53.9 | 56.5 | 83.0 | 67.0 | 96.2 | 61.6 | 68.6 | 69.4 |
| | | X | 86.0 | 87.6 | 70.9 | 23.1 | 52.8 | 47.5 | 80.3 | 61.5 | 93.6 | 59.8 | 63.9 | 66.1 |
| | L/14 | ✓ | 93.7 | 95.0 | 82.5 | 47.6 | 62.7 | 68.0 | 92.3 | 74.2 | 97.6 | 75.0 | 78.9 | 78.9 |
| | | X | 93.0 | 94.0 | 82.0 | 44.9 | 61.2 | 66.6 | 91.8 | 71.7 | 91.6 | 73.1 | 77.0 | 77.0 |
| Sig LIP | B/16 | ✓ | 90.7 | 94.1 | 85.8 | 43.9 | 62.0 | 42.3 | 89.2 | 69.6 | 97.4 | 74.9 | 75.7 | 75.1 |
| | | X | 86.3 | 90.4 | 69.5 | 35.1 | 58.6 | 32.5 | 84.6 | 55.9 | 89.5 | 64.8 | 62.1 | 66.3 |

As in the image-to-image retrieval experiments detailed in Sec. 5.1, we consider two strategies. In the former, we use the query text features to retrieve from the gallery. In the latter, we approach the task inter-modally by applying OVI to each query to obtain the complementary features.

**Results.** Table 2 (left) shows the results. Analogously to image-to-image retrieval, using the OVI-inverted features outperforms the intra-modal baseline. Specifically, the absolute performance gains range from 1% to 5% across all datasets, backbones, and VLMs. This outcome proves that intra-modal similarity comparisons lead to suboptimal performance independently from the considered modality. Hence, the intra-modal misalignment phenomenon involves both the image and text embedding spaces.

## 5.3    ZERO-SHOT IMAGE CLASSIFICATION

We evaluate the performance of modality inversion on inter-modal tasks, such as zero-shot image classification and image-text retrieval. We expect that transforming inter-modal tasks to intra-modal ones *hinders* performance due to intra-modal misalignment. Here we consider zero-shot image classification, while we report experiments on image-text retrieval in the supplementary material.

**Experiment design.** CLIP-like models perform zero-shot image classification by predicting the output class based on the similarity between the input image and a set of textual prompts in the form of "*a photo of a [CLASS]*", where CLASS represents each class name, such as "cat" or "dog". Following Zhou et al. (2022b), we take into account 11 datasets (see the supplementary material for more details). We consider three strategies. The first is the inter-modal baseline, which compares the features of the input image and the set of prompts. In the second, we apply OTI to the input image. In the third, OVI is applied to each textual prompt.

**Results.** In Tab. 2 (right) we report the performance of the first two strategies described above. Results for the third strategy are given in Appendix G. As expected, using modality inversion consistently leads to performance degradation across different VLMs and backbones. Note that the datasets used in zero-shot image classification are also employed for image-to-image retrieval in Sec. 5.1. This allows us to reuse the *same* OTI-inverted features for both tasks. Interestingly, the results are opposite: performance improves in image-to-image retrieval but decreases in zero-shot image classification. This contrast arises because, in the former, we transform an intra-modal task into an inter-modal one, while in the latter, we do the reverse. This experiment demonstrates that modality inversion does not inherently improve performance, as the same OTI-inverted features can either enhance or hinder results depending on the nature of the task.

## 5.4    ANALYZING MODALITY INVERSION

In this section we study how and why transforming native modality features into complementary ones via modality inversion leads to performance improvement on intra-modal tasks. For brevity, we consider only OTI, but we find that the same considerations apply to OVI. We consider the Cars dataset (Krause et al., 2013) and the CLIP ViT-B/32 model.

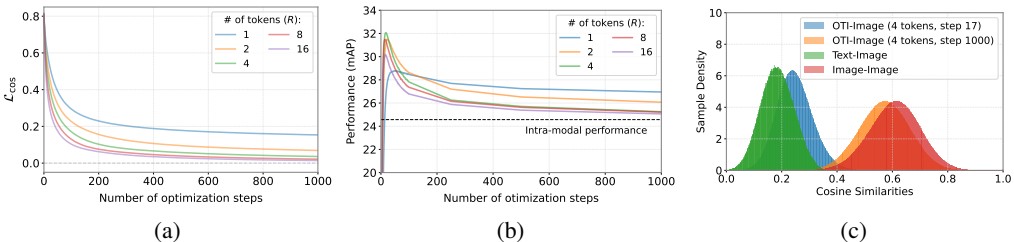

Figure 2: **(a, b)** Loss values and retrieval performance over OTI optimization steps for different numbers of pseudo-tokens $R$. **(c)** Distribution of pairwise image-image, text-image, and OTI-image cosine similarities. We consider the OTI-inverted features using four pseudo-tokens ($R = 4$) at two distinct optimization steps: the *performance peak* (step 17) and the *final step* (step 1000).

In Figs. 2(a) and 2(b) we investigate how the values of the loss $\mathcal{L}_{\cos}$ and the image-to-image retrieval performance vary based on the number of optimization steps and pseudo-tokens $R$. First, we notice that with a single pseudo-token (*i.e.* $R = 1$) the loss does not reach zero within a reasonable number of optimization steps. Conversely, as $R$ increases (*i.e.* the number of trainable parameters grows) the loss decreases more rapidly and approaches zero. As discussed in Sec. 4.3, as the loss decreases the OTI-inverted features shift away from the text manifold towards the image manifold, approaching the original native image features. This phenomenon is reflected in the image retrieval performance shown in Fig. 2(b), since for enough optimization steps and pseudo-tokens the performance approaches those obtained by the native image features. Moreover, we observe that, regardless of the value of $R$, the best performance corresponds to a relatively low number of optimization steps.

We argue that, in proximity to the performance peak observed during the optimization process, the OTI-inverted features capture the informative content of the corresponding image while retaining the inter-modal alignment. To support this claim, we compute cosine similarities for image-image, text-image, and OTI-image feature pairs. Specifically, we analyze the OTI-inverted features using four pseudo-tokens ($R = 4$) at two distinct optimization steps: the *performance peak* (step 17) and the *final step* (step 1000), where performance approaches the intra-modal baseline. In Fig. 2(c) we plot the distribution of these pairwise similarities. We observe that, at the performance peak, the similarity distribution of OTI-image matches the text-image similarity distribution, while at the final step it aligns with the image-image similarity distribution, confirming the drift of the OTI-inverted features toward the image manifold. This suggests that OTI-inverted features perform best when aligned with image features in the same way as text features, confirming our hypothesis that the performance improvement obtained by OTI stems from leveraging CLIP's inter-modal alignment.

Finally, we notice that $R = 1$ is *not* the optimal choice to achieve the best performance when using OTI. Still, we use $R = 1$ in the experiments as the associated OTI-inverted features are less prone to drift towards the native image features, thus being more robust to the number of optimization steps. Moreover, the main objective of this work is *not* to achieve the best results on the downstream tasks but rather show that using VLMs intra-modally is suboptimal.

## 5.5 THE ROLE OF INTRA-MODAL CONSTRAINTS

We investigate whether incorporating an intra-modal loss term during image-text contrastive pre-training effectively mitigates the issue of intra-modal misalignment. To this end, we consider SLIP (Mu et al., 2022), which adds a self-supervised intra-modal loss based on SimCLR (Chen et al., 2020) to the standard CLIP inter-modal contrastive loss $\mathcal{L}_{\text{CLIP}}$ (see Appendix C for more details). Such intra-modal loss encourages the model to produce similar representations for two augmentations of the same image, aiming to improve the intra-modal alignment within the image embedding space.

To verify this, we perform an image-to-image retrieval experiment following the evaluation protocol from Sec. 5.1. We report the results in Tab. 3. Notably, the OTI-inverted features achieve comparable performance to the native image ones. This contrasts with results from VLMs trained solely with an inter-modal contrastive loss (see Tab. 1), in which OTI led to a substantial performance boost. This experiment proves that SLIP's intra-modal loss effectively reduces intra-modal misalignment and suggests the importance of including such a loss when pre-training VLMs.

Table 3: Performance (mAP) evaluation on the image-to-image retrieval task using SLIP model. Blue rows indicate the usage of OTI-inverted features, while white rows refer to the intra-modal baseline. ✓ and ✗ denote inter-modal and intra-modal approaches, respectively.

| | Backbone | Inter modal | CUB | SOP | $\mathcal{R}$Oxford | $\mathcal{R}$Paris | Cars | Pets | Flowers | Aircraft | DTD | EuroSAT | Food101 | SUN397 | Caltech | UCF101 | ImageNet | *Average* |
|---|---|---|---|---|---|---|---|---|---|---|---|---|---|---|---|---|---|---|
| SLIP | B/16 | ✗ | **16.6** | **49.3** | 36.2 | 78.9 | 4.9 | 17.8 | **65.2** | **9.1** | 29.8 | **53.0** | 19.3 | 26.2 | 65.5 | 40.3 | 14.5 | 35.1 |
| | | ✓ | 16.2 | 48.8 | **36.4** | **79.3** | **5.0** | **19.3** | 65.1 | 9.0 | **30.5** | 50.6 | **20.0** | **26.4** | **67.6** | **40.6** | **14.8** | **35.3** |
| | L/16 | ✗ | 19.5 | **46.4** | 36.3 | **75.3** | 5.3 | 21.7 | 69.2 | 9.7 | 28.8 | **56.5** | 24.25 | 27.4 | 71.0 | 41.2 | 17.4 | 36.7 |
| | | ✓ | **19.6** | 45.8 | **38.0** | 75.1 | **5.5** | **23.3** | **70.2** | **9.8** | **29.7** | 53.7 | **25.2** | **27.8** | **72.3** | **41.4** | **18.2** | **37.1** |

## 5.6 THE ROLE OF THE MODALITY GAP

During CLIP pre-training, the temperature parameter $\tau$ in Eq. (1) critically affects the modality gap: higher temperatures considerably reduce or close it (Liang et al., 2022). To examine the impact of the modality gap on the intra-modal misalignment, we fine-tune a CLIP ViT-B/32 model on the COCO dataset (Lin et al., 2014) using a temperature $\tau = 1.0$, which closes the modality gap. As a reference, we repeat the experiment with $\tau = 0.01$, *i.e.* the value employed during CLIP pre-training. See Tab. A2 for more details on the magnitudes of the modality gap for the different models.

Table 4: Impact of the modality gap on the performance (mAP) for the image-to-image retrieval task on image retrieval datasets. Blue rows indicate the usage of OTI-inverted features, while white rows refer to the intra-modal baseline. ✓ and ✗ denote inter-modal and intra-modal approaches, respectively.

| Temperature | Inter modal | CUB | SOP | $\mathcal{R}$Oxford | $\mathcal{R}$Paris | Cars | *Average* |
|---|---|---|---|---|---|---|---|
| $\tau = 1$ *(no gap)* | ✗ | 15.9 | 23.7 | 29.3 | 46.6 | 19.3 | 27.0 |
| | ✓ | 14.0 | 20.4 | 26.7 | 43.1 | 17.4 | 24.2 |
| $\tau = 0.01$ | ✗ | 24.0 | 35.0 | 43.1 | 68.6 | 25.7 | 39.3 |
| | ✓ | **24.1** | **35.2** | **44.0** | **70.2** | **27.6** | **40.2** |

We reproduce our image-to-image retrieval experiments using these fine-tuned models and report results in Tab. 4. In the absence of the modality gap ($\tau = 1$) tackling intra-modal tasks inter-modally does not improve performance. This shows that closing the modality gap reduces the intra-modal misalignment. The results of the reference model ($\tau = 0.01$) prove that this outcome does not stem from the fine-tuning strategy. As also observed by Liang et al. (2022), we note that using higher temperature values during training leads to an overall performance decrease in downstream tasks, despite reducing the modality gap. For this reason, we argue that – in practice – simply increasing the temperature value in Eq. (1) does not represent a viable strategy to address intra-modal misalignment.

## 6 CONCLUSIONS

In this work we show that relying on intra-modal similarities computed with off-the-shelf VLMs is suboptimal for intra-modal tasks like image-to-image and text-to-text retrieval. This stems from the inter-modal contrastive loss employed for pre-training these models that leads to a modality gap and *intra-modal misalignment*. We propose to transform intra-modal tasks to inter-modal ones via two single-feature level modality inversion techniques. We demonstrate that this strategy improves performance as it exploits the inter-modal alignment of VLMs. Finally, we show that employing an intra-modal loss component during VLM pre-training or reducing the modality gap alleviates the impact of intra-modal misalignment.

**Limitations.** Our analyses demonstrate the significance of intra-modal misalignment when exploiting pre-trained CLIP models, but fall short of offering practical alternatives. The modality inversion techniques we propose are computationally expensive. They are based on iterative optimization of learnable input parameters (150 optimization steps for OTI and 1000 for OVI in our experiments). This limits their practical applicability and future work should concentrate on efficient methods to mitigate the *intra-modal misalignment*.

ACKNOWLEDGMENTS

This work was supported by funding from the Italian national project *Collaborative Explainable neuro-symbolic AI for Decision Support Assistant, CAI4DSA, CUP B13C23005640006.*

REPRODUCIBILITY STATEMENT

We have taken steps in this work to ensure the reproducibility of our results. All models and datasets used in our experiments are publicly available and we release the complete source code. In the main paper and appendices material we provide complete details of all experimental setups, including model architectures, training and evaluation protocols, and hyperparameters. All random seeds are fixed ensuring that others can replicate our results with the provided code. We believe that the measures we have taken to ensure reproducibility will facilitate straightforward replication and verification of our findings, as well as allow the community to build upon our results in the future.

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

---

**Algorithm 1** Optimization-based Textual Inversion (OTI)

---

1: **Input:** Image $I$, number of pseudo-tokens $R$, number of optimization steps $S$
2: Initialize $v^* = \{v_1^*, v_2^*, \ldots, v_R^*\}$
3: Extract image features: $\psi_I = f_\theta(I)$
4: **for** $s = 1$ to $S$ **do**
5:     Form $\overline{Y}_{v^*} = [E_v(\text{"a photo of"}), v^*]$
6:     Extract text features: $\psi_T = g_\phi(\overline{Y}_{v^*})$
7:     Compute loss: $\mathcal{L}_{\cos} = 1 - \cos(\psi_I, \psi_T)$
8:     Update $v^*$ to minimize $\mathcal{L}_{\cos}$
9: **end for**
10: **Output:** OTI-inverted features $\psi_T = g_\phi(\overline{Y}_{v^*})$

---

**Algorithm 2** Optimization-based Visual Inversion (OVI)

---

1: **Input:** Text $Y$, number of pseudo-patches $P$, number of optimization steps $S$
2: Initialize $w^* = \{w_1^*, w_2^*, \ldots, w_P^*\}$
3: Extract text features: $\psi_T = g_\phi(E_v(Y))$
4: **for** $s = 1$ to $S$ **do**
5:     Form input $\bar{I}_{w^*}$ using Eq. (3)
6:     Extract image features: $\psi_I = f_\theta(\bar{I}_{w^*})$
7:     Compute loss: $\mathcal{L}_{\cos} = 1 - \cos(\psi_I, \psi_T)$
8:     Update $w^*$ to minimize $\mathcal{L}_{\cos}$
9: **end for**
10: **Output:** OVI-inverted features $\psi_I = f_\theta(\bar{I}_{w^*})$

---

**Algorithms 1 and 2**. **Left**: OTI maps an image into the textual embedding space by optimizing pseudo-tokens. **Right**: OVI maps a text into the visual embedding space by optimizing pseudo-patches. Both approaches iteratively minimize the cosine distance between the feature representations of the native and complementary modality.

## APPENDIX A IMPLEMENTATION DETAILS

**OTI and OVI.** We report the pseudo-code of Optimization-based Textual Inversion (OTI) and Optimization-based Visual Inversion (OVI) in Algorithm 1 and Algorithm 2, respectively. Unless stated otherwise, we use the same hyperparameters for OTI and OVI. We employ the AdamW Loshchilov & Hutter (2019) optimizer with learning rate equal to 0.02, $\beta_1 = 0.9$, $\beta_2 = 0.999$, and weight decay 0.01. We perform 150 optimization steps for OTI and 1000 steps for OVI. For OTI, we consistently use a single pseudo-token ($R = 1$). In contrast, for OVI, we employ a number of pseudo-patches $P$ ranging from 1 to 4, depending on the considered model (see Sec. D for more details). On average, when using the CLIP ViT/B-32 model, OTI takes approximately 0.2 seconds per image, while OVI takes around 0.5 seconds per text prompt on a single A100 GPU (40GBs) with a batch size of 2048. The memory usage scales linearly with the batch size. Specifically, when using the CLIP ViT-B/32 model, OTI requires approximately 1,878 MiB plus 18.6 MiB per sample in the batch. For example, with a batch size of 128, the memory consumption is about 4,260 MiB. For OVI, the memory usage is approximately 2,218 MiB plus 16.2 MiB per sample, resulting in about 4,290 MiB with the same batch size. We use mixed precision to save memory and increase computational efficiency. In downstream tasks all the features are normalized to have a unit $L_2$-norm.

**CLIP fine-tuning.** To investigate the role of the modality gap on the intra-modal misalignment, we perform a fine-tuning of the CLIP model using different loss temperatures (see Sec. 5.6). In particular, we fine-tune the CLIP ViT B/32 model on the COCO training set for 30k steps, using a batch size of 512 and a learning rate of 1e-6. As an optimizer we employ AdamW with $\beta_1 = 0.9$, $\beta_2 = 0.98$ and a weight decay of 0.2. To mitigate possible overfitting issues, we train only the final projection layers. We train two different models, in the first we set the loss temperature parameter $\tau = 1.0$ (first two rows of Tab. 4), while in the second we use $\tau = 0.01$ (last two rows of Tab. 4).

## APPENDIX B MORE INSIGHTS ON INTRA-MODAL MISALIGNMENT

To provide quantitative insights into the intra-modal misalignment issue we conduct a simple experiment using the CLIP ViT-B/32 model and the "Dogs vs Cats" dataset (Elson et al., 2007). This dataset consists of 25K images evenly distributed between two classes: dog and cat. Our goal is to demonstrate that, despite inter-modal alignment, the intra-modal similarity scores are misaligned, *i.e.* they might not reflect those of actual images and texts, as illustrated in the left section of Fig. 1.

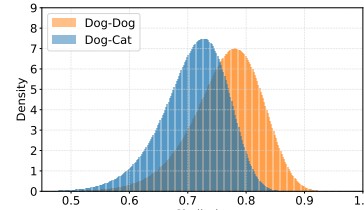

Figure A1: Distribution of pairwise dog-dog and dog-cat image similarities. Overlap highlights the intra-modal misalignment issue.

We start by filtering out images with incorrect inter-modal alignment to class-specific prompts. Specifically, we remove dog images that exhibit higher similarity to the prompt "a photo of a cat" than to the prompt "a photo of a dog". Then

we use the dog-related prompt to query the gallery of all images and filter out the minimal number of images that are incorrectly ranked for this query. We repeat the same procedure for cat images. This filtering ensures perfect inter-modal alignment and text-image retrieval scores.

On the resulting filtered dataset, we perform image-to-image retrieval using dog images as queries and the whole set of images as the gallery. If inter-modal alignment guarantees intra-modal alignment, all dog images should rank higher than cat images for any dog query, resulting in perfect retrieval. However, our results contradict this assumption. Specifically, we observe a mean Average Precision (mAP) of 83.1% and an average R-Precision of 73.2%, where R-Precision represents the precision at rank R, with R being the total number of relevant items for a given query. These findings indicate that on average at least 26.8% of dog images are ranked below cat images for a given dog query. Figure A1 qualitatively illustrates this issue, revealing significant overlap between the distributions of pairwise dog-dog and dog-cat image similarities. We observe similar results when employing cat images as queries. Given the evidence of intra-modal misalignment in such a toy dataset, we believe that the issue is likely to be even more pronounced in more complex datasets with more classes.

## APPENDIX C  ADDITIONAL VLMS

In this section, we provide a more detailed explanation of the SigLIP and SLIP models, highlighting their key differences from CLIP.

**SigLIP.** In SigLIP (Zhai et al., 2023), given a batch of image-caption pairs $B = \{(I_i, Y_i)\}_{i=1}^N$, training maximizes the cosine similarity for the $N$ correct pairs and minimizes it for the $N^2 - N$ incorrect pairs. Unlike the softmax-based contrastive loss from Eq. (1) used in CLIP, SigLIP employs a sigmoid-based loss that avoids global normalization factors. Each image-text pair is processed independently, transforming the learning task into a binary classification problem across all pair combinations. The matching pair $(I_i, Y_i)$ receives a positive label, while all other pairs $(I_i, Y_{j \neq i})$ receive negative labels. SigLIP consists of an image encoder $f_\theta$ and a text encoder $g_\phi$. We denote the image and text embeddings as $\psi_I^i = f_\theta(I_i)$ and $\psi_T^i = g_\phi(Y_i)$, respectively. The loss employed by SigLIP is:

$$\mathcal{L}_{\text{SigLIP}} = -\frac{1}{N} \sum_{i=1}^N \sum_{j=1}^N \log \left( \frac{1}{1 + e^{z_{ij}(-c(\psi_I^i, \psi_T^j)/\tau + b)}} \right), \tag{4}$$

where $c(\cdot, \cdot)$ denotes the cosine similarity, $\tau$ is a learnable temperature parameter, $b$ is a learnable bias, and $z_{ij}$ is the label for a given image and text input ($z_{ij} = 1$ if $i = j$ and $z_{ij} = -1$ otherwise). Similar to CLIP, SigLIP's loss does not include explicit intra-modal constraints; the loss focuses solely on inter-modal alignment between image and text embeddings, without directly enforcing intra-modal alignment.

**SLIP.** SLIP (Mu et al., 2022) is a VLM trained with both language supervision and image self-supervision. Its loss function combines two components: the inter-modal loss used in CLIP (Eq. (1)), and an intra-modal self-supervised term. For the self-supervised component, SLIP uses an adaptation of SimCLR (Chen et al., 2020). This loss, denoted as $\mathcal{L}_{\text{SimCLR}}$, maximizes features similarities between two different views (*i.e.* augmentation) of the same image:

$$\mathcal{L}_{\text{SimCLR}} = -\frac{1}{2N} \sum_{i=1}^{2N} \log \frac{\exp \left( c(\psi_I^{p(i)}, \psi_I^{q(i)})/\tau \right)}{\sum_{k=1, k \neq i}^{2N} \exp \left( c(\psi_I^i, \psi_I^k)/\tau \right)}, \tag{5}$$

where $\psi_I^j$ indicates the image embedding of a sample $j$, $p(i)$ and $q(i)$ are two augmented views of the image $i$, $c(\cdot, \cdot)$ denotes the cosine similarity, and $\tau$ is a temperature parameter.

The final loss used in SLIP is a combination of CLIP and self-supervised losses:

$$\mathcal{L}_{\text{SLIP}} = \mathcal{L}_{\text{CLIP}} + \mathcal{L}_{\text{SimCLR}}. \tag{6}$$

By incorporating the intra-modal self-supervised loss, SLIP encourages better intra-modal alignment within the image embedding space. We confirm this empirically in Tab. 3.

## APPENDIX D    SELECTING THE NUMBER OF PSEUDO-PATCHES FOR OVI

In Sec. 4.3, we observe that for certain experiments a single pseudo-patch (*i.e.* $P = 1$) is not enough to encapsulate the informative content of the associated text. To determine the optimal number of pseudo-patches for each VLM, we conducted a text-to-text retrieval experiment on the Flickr30K (Plummer et al., 2015) validation set, varying the number of $P$ from 1 to 16.

Table A1 presents the results of this ablation. We observe that the ideal number of pseudo-patches changes depending on the model. In particular, larger VLMs – with a greater number of input patches $U$ – tend to require more pseudo-patches. We hypothesize that this is because, as the number of patches increases, the influence of a single pseudo-patch decreases, necessitating a larger number to capture sufficient information.

Table A1: Ablation on the number of OVI pseudo-patches for text-to-text retrieval on the Flickr30K validation set. The highest mAP score in each row is highlighted in bold, with the corresponding value of $P$ representing the number of pseudo-patches used in the experiments.

| VLM | Backbone | Intra-modal | Number of Pseudo-Patches $P$ | | | | |
|-----|----------|-------------|------|------|------|------|------|
| | | | 1 | 2 | 4 | 8 | 16 |
| CLIP | B/32 | 51.4 | **54.5** | 52.9 | 51.8 | 51.6 | 51.6 |
| | L/14 | 52.6 | 51.7 | 55.2 | **56.0** | 55.3 | 54.1 |
| OPEN | B/32 | 57.3 | **59.6** | 57.9 | 57.5 | 57.4 | 57.4 |
| | L/14 | 59.6 | 60.6 | **62.5** | 62.4 | 61.2 | 60.4 |
| SigLIP | B/16 | 56.3 | 45.2 | 58.0 | **60.1** | 59.9 | 59.4 |
| SLIP | B/16 | 45.8 | **46.4** | 46.4 | 46.1 | 45.9 | 45.9 |
| | L/16 | 49.8 | 48.9 | **50.0** | 49.8 | 49.9 | 49.8 |

## APPENDIX E    DIFFERENT VLM, DIFFERENT MODALITY GAP

In Sec. 5.6 we show that in the absence of the modality gap, tackling intra-modal tasks inter-modally does not improve performance. This demonstrates that closing the modality gap helps reduce intra-modal misalignment.

Following Liang et al. (2022) we define the modality gap as the difference between the two centroids of the image and text modality embeddings:

$$\Delta_{\text{gap}} = \frac{1}{N} \sum_{i=1}^{N} x_i - \frac{1}{N} \sum_{i=1}^{N} y_i, \qquad (7)$$

where $x_i$ and $y_i$ are the L2-normalized image and text embeddings, and $N$ is the number of pairs.

To facilitate a clearer comparison across different VLMs, in Table A2 we report the magnitude of the modality gaps evaluated on the COCO validation split. We observe that integrating an intra-modal constraint (*e.g.* SLIP) or using a higher temperature in the contrastive loss (*e.g.* our fine-tuned model with temperature $\tau = 1$) helps reduce or even eliminate the modality gap. By analyzing Tabs. 1 to 4, we confirm our hypothesis that exists a positive correlation between the magnitude of the modality gap and the improvement in approaching intra-modal tasks inter-modally using OTI (or OVI).

Table A2: $\|\Delta_{\text{gap}}\|$ for different VLMs on COCO.

| VLM | Backbone | Loss | $\|\Delta_{\text{gap}}\|$ |
|-----|----------|------|------|
| CLIP | B/32 | $\mathcal{L}_{\text{CLIP}}$ | 0.82 |
| | L/14 | | 0.82 |
| OPEN | B/32 | $\mathcal{L}_{\text{CLIP}}$ | 0.82 |
| | L/14 | | 0.80 |
| SigLIP | B/16 | $\mathcal{L}_{\text{SigLIP}}$ | **1.05** |
| SLIP | B16 | $\mathcal{L}_{\text{CLIP}} + \mathcal{L}_{\text{SimCLR}}$ | 0.57 |
| | L/16 | | 0.49 |
| Fine-tuned CLIP | B/32 | $\mathcal{L}_{\text{CLIP}}(\tau = 1)$ | **0.007** |
| | | $\mathcal{L}_{\text{CLIP}}(\tau = 0.01)$ | 0.88 |

## APPENDIX F    DATASET DETAILS

Our experimental evaluation is performed on 18 datasets. Here we report all the evaluated splits and details of the datasets used in our experiments.

**Zero-shot Image Classification.** Following Zhou et al. (2022b), we validate our zero-shot image classification experiments on 11 publicly available datasets with diverse characteristics: ImageNet (Deng et al., 2009) for large-scale object classification; Caltech101 (Fei-Fei et al., 2004) for general object classification; EuroSAT (Helber et al., 2019) for satellite image recognition;

Food101 (Bossard et al., 2014), FGVCAircraft (Maji et al., 2013), OxfordPets (Parkhi et al., 2012), Flowers102 (Nilsback & Zisserman, 2008), and StanfordCars (Krause et al., 2013) for fine-grained classification; UCF101 (Soomro et al., 2012) for action recognition; and the Describable Textures Dataset (DTD) (Cimpoi et al., 2014) for texture classification. Following Zhou et al. (2022b), we discard the "BACKGROUND Google" and "Faces easy" classes from Caltech101. For UCF101 – a video dataset – we follow Radford et al. (2021) and use the middle frame of each video clip as the input image. In all classification experiments, we report the accuracy results on the test set.

**Image-to-Image Retrieval.** For image-to-image retrieval experiments, we use the 11 datasets also employed for zero-shot image classification and four widely used datasets commonly used for metric learning and image retrieval: CUB-200-2011 (CUB) (Wah et al., 2011), Stanford Online Products (SOP) (Oh Song et al., 2016), $\mathcal{R}$Oxford (Radenović et al., 2018), and $\mathcal{R}$Paris (Radenović et al., 2018), for a total of 15 datasets. In the 11 datasets used for zero-shot image classification, we use the test set as the query set and the training set as the gallery. For CUB, the entire dataset is used as both the query and gallery sets. In SOP, both the query and gallery sets are taken from the test set.

In all experiments involving $\mathcal{R}$Oxford and $\mathcal{R}$Paris, we follow the standard benchmark and include the $\mathcal{R}$1M distractor set, containing 1 million images, as negative samples for all the queries. For brevity in the paper we report only the metric calculated on the Easy setting Radenović et al. (2018). For image-to-image retrieval evaluation, we use the standard mean Average Precision (mAP) metric. Importantly, by using the same 11 datasets employed for zero-shot classification, we can evaluate the performance of the *same* OTI-inverted features on both inter-modal zero-shot classification and intra-modal image retrieval tasks

**Text-to-Text Retrieval.** We performed our text-to-text retrieval experiments using three image-caption datasets: COCO (Lin et al., 2014), Flickr30K (Plummer et al., 2015), and nocaps (Agrawal et al., 2019). We selected these datasets for two reasons: they contain short, descriptive text similar to the ones used to train VLMs, and they provide multiple captions for each image. In our evaluation, we use the first caption of each image as the query and aim to retrieve the other captions associated with the same image from a gallery of all captions in the dataset. On average, COCO and Flickr30K images have 5 captions each, while nocaps images have 10. We use the Karpathy split (Karpathy & Fei-Fei, 2015) for both COCO and Flickr30K and report results using captions from the test split. For nocaps, we report results on the validation split. Although these datasets contain images associated with captions, we ignore the images in this setting. We use mAP as the evaluation metric similar to the image retrieval experiments.

## APPENDIX G    ADDITIONAL EXPERIMENTS

Here we report additional experiments to support our claims about the importance of approaching tasks inter-modally when using constrastively trained VLMs.

**Zero-shot Image Classification with OVI.** Due to space limitation, in Sec. 5.3, we provide a brief overview of how CLIP-like models can perform zero-shot image classification. Here, we offer a more detailed explanation for clarity. Given an image $I$ and a set of textual prompts $\mathcal{Y} = \{Y_i \,|\, i = 1, \ldots, C\}$, where $C$ is the number of classes, each text prompt $Y_i$ is formatted as: "a photo of a [CLASS$_i$]", with CLASS$_i$ representing a specific class name, such as "cat" or "dog". Let the image features be denoted as $\psi_I = f_\theta(I)$ and the text features for each prompt by $\psi_T^i = g_\phi(Y_i)$. The probability of the image belonging to each class is then given by:

$$p(y = i|I) = \frac{\exp(c(\psi_T^i, \psi_I)/\tau)}{\sum_{j=1}^{C} \exp(c(\psi_T^j, \psi_I)/\tau)}, \tag{8}$$

where $c(\cdot, \cdot)$ denotes cosine similarity and $\tau$ is a temperature parameter.

In Sec. 5.3, we transform zero-shot image classification from being natively inter-modal to intra-modal by applying OTI to the input image $I$ (see the right section of Tab. 2). As expected, this consistently leads to performance degradation for different VLMs and backbones, demonstrating that modality inversion does not inherently improve performance. Similarly, we perform an experiment where we approach the zero-shot classification task *intra-modally* by applying OVI to each textual prompt, with results reported in Tab. A3. Consistent with our previous findings, we observe that approaching classification intra-modally hinders the performance.

Table A3: Performance (accuracy) evaluation on the zero-shot image classification task. Purple rows indicate the usage of OVI-inverted features, while white rows refer to the inter-modal baseline. ✓ and X denote inter-modal and intra-modal approaches, respectively.

| | Backbone | Inter modal | Cars | Pets | Flowers | Aircraft | DTD | EuroSAT | Food101 | SUN397 | Caltech | UCF101 | ImageNet | Average |
|---|---|---|---|---|---|---|---|---|---|---|---|---|---|---|
| CLIP | B/32 | ✓ | 60.4 | 87.5 | 67.0 | 19.1 | 43.6 | 45.2 | 80.5 | 62.0 | 91.2 | 62.0 | 62.1 | 61.9 |
| | | X | 37.4 | 59.9 | 35.0 | 9.2 | 26.2 | 18.9 | 65.1 | 44.1 | 83.9 | 51.2 | 42.9 | 43.1 |
| | L/14 | ✓ | 76.8 | 93.6 | 79.3 | 32.5 | 53.0 | 58.1 | 91.0 | 67.6 | 94.9 | 74.2 | 73.5 | 72.2 |
| | | X | 46.9 | 71.1 | 65.1 | 23.3 | 41.4 | 23.8 | 73.6 | 46.2 | 41.6 | 63.5 | 54.8 | 50.1 |
| OPEN | B/32 | ✓ | 88.4 | 90.3 | 73.5 | 24.4 | 53.9 | 56.5 | 83.0 | 67.0 | 96.2 | 61.6 | 68.6 | 69.4 |
| | | X | 81.4 | 82.1 | 62.4 | 17.9 | 45.8 | 36.6 | 76.1 | 56.9 | 93.6 | 55.1 | 59.6 | 60.7 |
| | L/14 | ✓ | 93.7 | 95.0 | 82.5 | 47.6 | 62.7 | 68.0 | 92.3 | 74.2 | 97.6 | 75.0 | 78.9 | 78.9 |
| | | X | 78.6 | 85.3 | 71.1 | 35.9 | 48.6 | 47.9 | 86.2 | 50.7 | 92.9 | 62.4 | 67.3 | 66.1 |
| SigLIP | B/16 | ✓ | 90.7 | 94.1 | 85.8 | 43.9 | 62.0 | 42.3 | 89.2 | 69.6 | 97.4 | 74.9 | 75.7 | 75.1 |
| | | X | 67.2 | 68.9 | 32.6 | 23.5 | 40.5 | 14.2 | 59.6 | 27.8 | 35.1 | 21.0 | 22.1 | 37.5 |

Table A4: Performance evaluation on the image-to-text and on the text-to-image retrieval task. Blue rows and Purple rows indicate the usage of OTI- and OVI-inverted features, respectively. White rows refer to the inter-modal baselines. ✓ and X denote inter-modal and intra-modal approaches, respectively.

| | | | Image-to-Text | | | | | | Text-to-Image | | | | | |
| | | | Flickr30k | | | COCO | | | Flickr30k | | | COCO | | |
| | Backbone | Inter modal | R@1 | R@5 | R@10 | R@1 | R@5 | R@10 | R@1 | R@5 | R@10 | R@1 | R@5 | R@10 |
|---|---|---|---|---|---|---|---|---|---|---|---|---|---|---|
| CLIP | B/32 | ✓ | 78.8 | 94.9 | 98.2 | 50.1 | 75.0 | 83.5 | 58.8 | 83.5 | 90.0 | 30.5 | 56.0 | 66.9 |
| | | X | 64.5 | 86.6 | 92.5 | 39.8 | 64.5 | 74.6 | 52.7 | 77.9 | 86.2 | 25.6 | 49.1 | 60.5 |
| | L/14 | ✓ | 85.2 | 97.4 | 99.2 | 56.3 | 79.3 | 86.6 | 64.9 | 87.3 | 92.0 | 36.5 | 61.0 | 71.1 |
| | | X | 75.8 | 92.9 | 95.9 | 49.0 | 72.8 | 81.2 | 60.7 | 84.8 | 90.3 | 33.2 | 55.1 | 67.7 |
| OPEN | B/32 | ✓ | 79.2 | 93.8 | 96.2 | 53.5 | 77.7 | 86.0 | 61.1 | 84.9 | 90.9 | 37.1 | 62.4 | 72.7 |
| | | X | 72.8 | 90.3 | 94.1 | 49.2 | 73.4 | 82.0 | 57.4 | 81.5 | 88.4 | 33.1 | 58.0 | 68.4 |
| | L/14 | ✓ | 89.1 | 98.6 | 99.7 | 63.3 | 84.2 | 90.4 | 73.4 | 91.8 | 95.5 | 45.7 | 70.1 | 79.2 |
| | | X | 86.0 | 97.7 | 98.9 | 60.8 | 81.5 | 88.3 | 67.4 | 88.1 | 93.0 | 39.0 | 63.4 | 73.2 |
| SigLIP | B/16 | ✓ | 89.0 | 98.0 | 99.2 | 65.7 | 85.4 | 91.2 | 74.6 | 92.3 | 95.6 | 47.8 | 72.4 | 81.0 |
| | | X | 81.8 | 95.5 | 97.3 | 57.0 | 79.0 | 86.2 | 57.9 | 82.6 | 88.7 | 33.7 | 58.2 | 68.9 |

**Image-Text Retrieval with OTI and OVI.** To provide additional experimental evidence that transforming inter-modal tasks in intra-modal ones hinders performance, we conduct an experiment on image-text retrieval using the COCO (Lin et al., 2014) and Flickr30K (Plummer et al., 2015) datasets. We use the Karpathy splits Karpathy & Fei-Fei (2015) for both datasets and report results on the test split. Following standard the evaluation benchmark, we report Recall@$K$ scores with $K = 1, 5$, and 10. Specifically, in image-to-text retrieval, we apply OTI to the query image, while in text-to-image retrieval, we apply OVI to the query text. We then compare the results with the inter-modal baseline, which directly compares image and text features. Results in Tab. A4 confirm our findings from the zero-shot image classification task: transforming an inter-modal task into an intra-modal one leads to performance degradation due to intra-modal misalignment.

**Text-to-text Retrieval on Purely Textual Datasets.** In Sec. 5.2 we conduct a text-to-text retrieval experiment using image captioning datasets to avoid a mismatch with VLMs pre-training data. In this section, we evaluate the performance of OVI on purely textual datasets using the CLIP ViT B/32 model. Specifically, we select seven datasets from the NanoBEIR benchmark[1] spanning diverse domains such as scientific documents (SciDOCS) and climate-related texts (ClimateFEVER). We discard Question-Answering (QA) datasets and those with queries whose average length exceeds the context length of CLIP's text encoder (77 tokens). Additionally, we include the IMDB Reviews (Maas et al., 2011) and the 20 Newsgroups (Lang, 1995) datasets.

---

[1]https://huggingface.co/collections/zeta-alpha-ai/nanobeir

Table A5: Performance (mAP) evaluation on the text-to-text retrieval task using purely textual datasets. Purple rows indicate the usage of OVI-inverted features. ✓ and ✗ denote inter-modal and intra-modal approaches, respectively.

| Method | Inter modal | IMDB | 20News. | Climate | DBPedia | FEVER | NFCorpus | NQ | SciDocs | SciFact | Average |
|---|---|---|---|---|---|---|---|---|---|---|---|
| Baseline | ✗ | 52.2 | 19.2 | 11.2 | 30.3 | 58.4 | 8.9 | 23.3 | 13.5 | 26.3 | 27.0 |
| OVI | ✓ | **52.3** | **33.1** | **15.3** | **39.1** | **70.5** | **12.2** | **33.6** | **16.8** | **33.2** | **34.0** |

Table A6: Performance (mAP) comparison between the proposed modality inversion techniques and the adapter-based approach on the image-to-image (**left**) and text-to-text (**right**) retrieval tasks. Blue rows and Purple rows indicate the usage of OTI- and OVI-inverted features, respectively. ✓ and ✗ denote inter-modal and intra-modal approaches, respectively.

| Method | Inter modal | CUB | SOP | ROxford | RParis | Cars | Average |
|---|---|---|---|---|---|---|---|
| Baseline | ✗ | 22.9 | 34.4 | 42.6 | 67.9 | 24.6 | 38.5 |
| Adapter | ✓ | 23.7 | 35.0 | **44.3** | 69.5 | 25.5 | 39.6 |
| OTI | ✓ | **24.6** | **35.1** | 43.0 | **70.3** | **28.0** | **40.2** |

| Method | Inter modal | Flickr30k | COCO | nocaps | Average |
|---|---|---|---|---|---|
| Baseline | ✗ | 51.7 | 26.2 | 35.1 | 37.7 |
| Adapter | ✓ | 51.9 | **28.3** | 37.8 | 39.3 |
| OVI | ✓ | **54.8** | **28.3** | **38.8** | **40.6** |

All selected datasets comprise texts that cannot be easily represented visually. Examples include "Learning Actionable Representations with Goal-Conditioned Policies" (SciDocs), "Atheism, philosophy, and the absence of belief in deities" (20 Newsgroup), and "The carbon footprint on wind energy is significant" (ClimateFEVER). Since gallery texts often exceed CLIP's context length, we employ the Llama-3.2-1B-Instruct[2] Large Language Model (Dubey et al., 2024) to summarize them to fit within the token limit.

We report the results in Tab. A5. OVI achieves a significant performance improvement over the intra-modal baseline. This outcome demonstrates that OVI is effective even when considering texts that can not be easily represented visually.

**From Intra-modal to Inter-modal via Adapters.** To broaden our comparative analysis we conduct an additional experiment where we train two single-layer linear adapters: one maps image features to text features (aligned with the goal of OTI), and the other maps text features to image features (aligned with the goal of OVI). For training, we leverage the LLaVA-CC3M[3] dataset (Liu et al., 2024), which comprises 595K image-text pairs. This dataset is derived by filtering the CC3M dataset (Sharma et al., 2018) to achieve a more balanced distribution of concept coverage. We train each adapter using a cosine loss that minimizes the distance between the adapter output and the corresponding complementary features. Additionally, following Patel et al. (2024), we also employ a CLIP-based contrastive loss component.

Table A6 presents the results for image-to-image and text-to-text retrieval tasks using the CLIP ViT-B/32 model. The adapter-based approach improves performance over the intra-modal baseline for both tasks. These findings support our claim that approaching the task *inter-modally* enhances performance thanks to CLIP's inherent inter-modal alignment. Interestingly, we observe that OTI and OVI outperform the adapter-based approach in most scenarios. This result emphasizes the effectiveness of OTI and OVI, as they do not require a training dataset but rather map individual features directly to the complementary modality without relying on external resources.

**From Intra-modal to Inter-modal via Captioning.** We compare the performance of OTI on image-to-image retrieval with a captioning-based approach. Specifically, given a query image, we first generate the caption using a pre-trained captioning model, then use CLIP's text encoder to extract the text features to perform retrieval.

---

[2]https://huggingface.co/meta-llama/Llama-3.2-1B-Instruct
[3]https://huggingface.co/datasets/liuhaotian/LLaVA-CC3M-Pretrain-595K

Table A7: Performance (mAP) comparison between the proposed OTI technique and the captioning-based approach on the image-to-image retrieval task. Blue rows indicate the usage of OTI-inverted features. ✓ and ✗ denote inter-modal and intra-modal approaches, respectively.

| Method | Inter modal | CUB | SOP | $\mathcal{R}$Oxford | $\mathcal{R}$Paris | Cars | *Average* |
|---|---|---|---|---|---|---|---|
| Baseline | ✗ | 22.9 | 34.4 | 42.6 | 67.9 | 24.6 | 38.5 |
| DeCap | ✓ | 4.4 | 2.0 | 0.1 | 1.2 | 2.5 | 2.0 |
| CoCa (COCO) | ✓ | 3.5 | 0.8 | 0.0 | 0.7 | 1.8 | 1.4 |
| CoCa (LAION) | ✓ | 17.6 | 3.9 | 8.4 | 28.2 | 23.6 | 16.3 |
| OTI | ✓ | **24.6** | **35.1** | **43.0** | **70.3** | **28.0** | **40.2** |

We experiment with three pre-trained captioning models: DeCap (Li et al., 2023), which directly generates captions from CLIP image features; CoCa (LAION)[4] (Yu et al.), trained on the Laion2B (Schuhmann et al., 2022) dataset; and CoCa (COCO)[5], pre-trained on Laion2B and fine-tuned on COCO (Lin et al., 2014).

Table A7 shows the results using the CLIP ViT-B/32 model. Regardless of the captioning model, the captioning-based approaches achieve unsatisfactory performance, even falling short of the intra-modal baseline despite leveraging CLIP's inter-modal alignment. This outcome stems from the fact that the generated captions are not discriminative enough to perform image retrieval. This is particularly evident in fine-grained domains such as the buildings of the $\mathcal{R}$Oxford and $\mathcal{R}$Paris datasets (Radenović et al., 2018). Figure A2 shows an example of generated captions for a randomly chosen image from the $\mathcal{R}$Oxford dataset. We observe that all the captioning models generate generic and not sufficiently discriminative captions. CoCa (LAION) produces a more precise description than the other models, reflecting its higher performance.

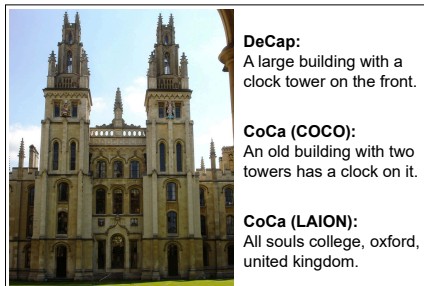

**DeCap:**
A large building with a clock tower on the front.

**CoCa (COCO):**
An old building with two towers has a clock on it.

**CoCa (LAION):**
All souls college, oxford, united kingdom.

Figure A2: Captions generated by pre-trained captioning models for an image from the $\mathcal{R}$Oxford dataset.

**Intra-OTI Similarity Comparisons.** To further support our claim that the performance improvement in image-to-image retrieval stems from CLIP's inter-modal alignment and not from the modality inversion itself, we perform an experiment where we apply OTI to both query and gallery images. Since OTI maps image features into text features, this intra-OTI strategy involves intra-modal similarity comparisons within the text embedding space.

Table A8 shows the results on image retrieval datasets using the CLIP ViT-B/32 model. We observe that employing inter-modal similarity comparisons by applying OTI only to the query images achieves better performance than using intra-modal similarities with the intra-OTI approach. This confirms that modality inversion techniques do not inherently improve performance. Instead, their effectiveness lies in leveraging CLIP's inter-modal alignment by transforming intra-modal tasks into inter-modal ones.

Table A8: Performance (mAP) evaluation on the image-to-image retrieval task. ✓ and ✗ denote inter-modal and intra-modal approaches, respectively.

| Method | Inter modal | CUB | SOP | $\mathcal{R}$Oxford | $\mathcal{R}$Paris | Cars | *Average* |
|---|---|---|---|---|---|---|---|
| Baseline | ✗ | 22.9 | 34.4 | 42.6 | 67.9 | 24.6 | 38.5 |
| Intra-OTI | ✗ | 21.3 | 31.9 | 42.3 | 68.2 | 24.9 | 37.7 |
| OTI (ours) | ✓ | **24.6** | **35.1** | **43.0** | **70.3** | **28.0** | **40.2** |

**Impact of the OTI Template Sentence.** As detailed in Sec. 4.1, for OTI we concatenate the template sentence "a photo of" with the pseudo-token $v^*$ to craft the prompt "a photo of $v^*$". To study the impact of the template sentence on the performance, we test the following prompts: 1) "an image of $v^*$"; 2) "we see $v^*$ in this photo"; and 3) "$v^*$" (the *empty prompt*).

---

[4]https://huggingface.co/laion/CoCa-ViT-B-32-laion2B

[5]https://huggingface.co/laion/mscoco_finetuned_CoCa-ViT-B-32

Table A9: Impact on the performance (mAP) of the OTI template sentence used in image-to-image retrieval. Each prompt is given by the combination of a template sentence with the pseudo-token $v^*$.

| OTI Prompt | CUB | SOP | $\mathcal{R}$Oxford | $\mathcal{R}$Paris | Cars | *Average* |
|---|---|---|---|---|---|---|
| "$v^*$" (*empty prompt*) | 24.0 | 34.6 | **43.7** | 69.6 | 28.2 | 40.0 |
| "We see $v^*$ in this photo" | 24.5 | 34.7 | 43.0 | 69.7 | **28.3** | 40.0 |
| "An image of $v^*$" | 24.0 | 34.8 | 43.1 | **70.7** | 28.3 | **40.2** |
| "A photo of $v^*$" (**ours**) | **24.6** | **35.1** | 43.0 | 70.3 | 28.0 | **40.2** |

Table A9 reports the image-to-image retrieval results using the CLIP ViT-B/32 model. We observe that all the considered prompts achieve comparable performance. These results demonstrate the robustness of the OTI technique to the template sentence.

**Combining Native and OTI-Inverted Features.** We conduct an experiment on image-to-image retrieval to assess whether combining native image features with the corresponding OTI-inverted features improves the performance. Let $\psi_I = f_\theta(I)$ be the native image features and $\psi_T = g_\phi(\overline{Y}_{v^*})$ be the OTI-inverted features. We query the gallery using a weighted combination of native and OTI-inverted representations:

$$\psi_{IT} = \alpha * \psi_T + (1 - \alpha) * \psi_I, \tag{9}$$

where $\alpha \in [0, 1]$ is a weighting factor that controls the contribution of each component.

Table A10 reports the results on image-to-image retrieval datasets for varying values of $\alpha$ using the CLIP ViT-B/32 model. Interestingly, for $\alpha$ large enough, combining native and inverted features obtains better results than relying solely on either of them. Notably, regardless of the $\alpha$ value, we observe that employing OTI-inverted features always improves the performance over the intra-modal baseline. We leave further investigation of the combination of native and OTI-inverted features to future work.

Table A10: Performance (mAP) evaluation of the combination between native and OTI-inverted features for varying weighting factors $\alpha$ for image-to-image retrieval.

| Method | CUB | SOP | $\mathcal{R}$Oxford | $\mathcal{R}$Paris | Cars | *Average* |
|---|---|---|---|---|---|---|
| Baseline ($\alpha = 0$) | 22.9 | 34.4 | 42.6 | 67.9 | 24.6 | 38.5 |
| OTI ($\alpha = 0.25$) | 24.0 | 35.6 | 44.9 | 70.1 | 25.9 | 40.1 |
| OTI ($\alpha = 0.50$) | 24.6 | **36.1** | **46.7** | 71.0 | 27.0 | 41.1 |
| OTI ($\alpha = 0.75$) | **24.8** | 35.9 | 46.3 | **71.1** | 27.7 | **41.2** |
| OTI ($\alpha = 1$) (**ours**) | 24.6 | 35.1 | 43.0 | 70.3 | **28.0** | 40.2 |

