# OpenReview forum: "Cross the Gap:  Exposing the Intra-modal Misalignment in CLIP via Modality Inversion"
_ICLR.cc/2025/Conference — ICLR 2025 Poster_

### Official Review · Reviewer_BNBn · 2024-10-30

**Soundness:** 4
**Presentation:** 4
**Contribution:** 3
**Rating:** 8
**Confidence:** 4

**Summary:**

This paper demonstrates that CLIP models, trained exclusively with inter-modal contrastive loss, perform suboptimally on intra-modal tasks like image-to-image and text-to-text retrieval. The authors attribute this limitation to a modality gap and the absence of intra-modal supervision during pre-training. To address this, they employ modality inversion techniques, transforming inputs across modalities to bridge this gap and enhance intra-modal performance. Through extensive experiments, they show that CLIP’s intra-modal performance improves by leveraging the structured nature of inter-modal cosine similarities, which are more consistent than intra-modal ones.

**Strengths:**

Well written, easy to follow.
Extensive experiments and analysis.

**Weaknesses:**

Lack. of alternative baselines, consider evaluating alternative baselines, such as using an image-captioning model to generate captions for query images, followed by image-to-image retrieval based on these captions. This could provide a comparative perspective on the effectiveness of the proposed modality inversion techniques.

Minor: Are the ticks and crosses correct in Table 2. Right. ?

**Questions:**

1. Does combining the native image representation with its inverted (cross-modal) image representation provide any performance benefits?

2. For text-to-text retrieval, have you tried the method on a purely textual task with less than 77 tokens context? Existing text retrieval datasets could be summarized to 77 tokens using an LLM. My assumption is that this approach would only be effective when the text can be represented visually, which might explain its success with text derived from image-captioning datasets.

3. Have you considered alternative baselines for image-to-image retrieval, such as using a captioning model?

4. Do you have any intuitions on why the image-image retrieval score for EuroSat dataset goes down with modality inversion?

---

> ### Author Response · Authors · 2024-11-20
> **Response to Reviewer BNBn (part 1).**
>
> We thank the Reviewer for their insightful review and for their appreciation of the clarity of our presentation and the our extensive experimental evaluation. Below we respond to each specific point raised by the Reviewer.
>
> > W1: Lack. of alternative baselines, consider evaluating alternative baselines, such as using an image-captioning model to generate captions for query images, followed by image-to-image retrieval based on these captions. This could provide a comparative perspective on the effectiveness of the proposed modality inversion techniques.
>
> We conducted additional experiments to address concerns about the lack of alternative baselines.
>
> #### Image Captioning Baseline
> Following the Reviewer’s suggestion, we investigated image-to-image retrieval performance using a captioning model to generate text descriptions of query images. Given a query image, we generated a caption using a pre-trained captioning model, extracted text features from the generated caption using the CLIP text encoder, and used these features to perform retrieval. We experimented with three captioning models:
> - DeCap [1], which directly generates captions from CLIP image features, making it the most comparable approach since OTI also relies only on CLIP image features;
> - [CoCa (LAION)](https://huggingface.co/laion/CoCa-ViT-B-32-laion2B-s13B-b90k) [2], trained on the Laion2B dataset; and
> - [CoCa (MSCOCO)](https://huggingface.co/laion/mscoco_finetuned_CoCa-ViT-B-32-laion2B-s13B-b90k) [2], pre-trained on Laion2B and fine-tuned on MSCOCO.
>
> The table below summarizes the image retrieval results in terms of mAP:
>
> | Method | Inter modal | CUB | SOP | ROxf. | RParis | Cars | AVG |
> |---------------|:----------:|:------:|:------:|:------:|:------:|:------:|:------:|
> | Intra-modal baseline | ❌ | 22.9 | 34.4 | 42.6 | 67.9 | 24.6 | 38.5 |
> | DeCap | ✅ | 4.4 | 2.0 | 0.1 | 1.2 | 2.5 | 2.0 |
> | CoCa (MSCOCO) | ✅ | 3.5 | 0.8 | 0.0 | 0.7 | 1.8 | 1.4 |
> | CoCa (LAION) | ✅ | 17.6 | 3.9 | 8.4 | 28.2 | 23.6 | 16.3 |
> | **OTI (ours)** | ✅ | **24.6** | **35.1** | **43.0** | **70.3** | **28.0** | **40.2** |
>
> In all cases captioning-based retrieval falls short of the performance of the intra-modal baseline, despite their leveraging of CLIP’s image-text alignment. Furthermore, the effectiveness of captions varies with the dataset domain: captioners struggle to produce discriminative captions for datasets featuring buildings (ROxford and RParis) while achieving better results in domains like cars (Cars). Inter-modal features derived via modality inversion (OTI), on the other hand, improve image retrieval performance on all datasets.
>
> To understand the variability in performance of different captioning models, we report captions generated by the three models for a randomly chosen image (“all_souls_000026”) from the ROxford dataset depicting the All Souls College:
>
> 1. DeCap: “a large building with a clock tower on the front.”;
> 2. CoCa (MSCOCO): “an old building with two towers has a clock on it.”; and
> 3. CoCa (LAION): “all souls college, oxford, united kingdom.”
>
> This example shows that the first two models fail to generate sufficiently discriminative captions, while CoCa (LAION) produces a more precise description, correlating with its higher performance among the captioning models.
>
> We hypothesize that a more advanced captioner, such as a large multimodal language model (e.g. ChatGPT-4V), could generate more precise descriptions and potentially improve retrieval performance by better leveraging cross-modal alignment. However, comparing such an approach with OTI, which does not rely on any external data, would not be fair.
>
> **(Continued in next message)**

---

> > ### Author Response · Authors · 2024-11-20
> > **Response to Reviewer BNBn (part 2).**
> >
> > #### Adapter Baselines
> > We performed additional experiments using adapters trained to map features from their native modality to the complimentary ones (as suggested by Reviewer pj7m). To train each adapter we used the [LLaVA-CC3M](https://huggingface.co/datasets/liuhaotian/LLaVA-CC3M-Pretrain-595K) dataset [3], which comprises 595K image-text pairs. Adapters are trained using a cosine loss to minimize the distance between the adapter output and the corresponding complementary features. Additionally, following Patel et al. [4], we incorporated a CLIP-based contrastive loss during training. We trained two separate adapters: one for mapping image features to text features (aligned with the goal of OTI) and another for mapping text features to image features (aligned with the goal of OVI).
> >
> > First for image-to-image retrieval:
> > |Method|Inter modal|CUB|SOP|ROxf.|RParis|Cars|AVG|
> > |--|:--:|:--:|:--:|:--:|:--:|:--:|:--:|
> > |Intra-modal baseline|❌|22.9|34.4|42.6|67.9|24.6|38.5|
> > |Adapter|✅|23.7|35.0|**44.3**|69.5|25.5|39.6|
> > |**OTI (ours)**|✅|**24.6**|**35.1**|43.0|**70.3**|**28.0**|**40.2**|
> >
> > And for text-text retrieval:
> > |Method|Inter modal|Flickr30k|COCO|nocaps|AVG|
> > |--|:--:|:--:|:--:|:--:|:--:|
> > |Intra-modal baseline|❌|51.7|26.2|35.1|37.7|
> > |Adapter|✅|51.9|**28.3**|37.8|39.3|
> > |**OVI (ours)**|✅|**54.8**|**28.3**|**38.8**|**40.6**|
> >
> > The adapter approach improves over the intra-modal baseline for both text and image retrieval tasks, which aligns with our hypothesis that leveraging inter-modal representations for intra-modal tasks enhances performance due to CLIP’s inherent inter-modal alignment. However, on average OTI and OVI outperform the adapter-based approach. This finding is particularly noteworthy given that OTI and OVI **do not require an additional training dataset**. Instead, they map individual features directly to the complementary modality without relying on external resources.
> >
> > > W2: Minor: Are the ticks and crosses correct in Table 2. Right. ?
> >
> > Yes, we confirm that ticks and crosses are correct in Table 2 (right) and apologize for the confusion. We acknowledge that the current formatting could be misleading and we are working on improving the clarity of the revised manuscript. In the current tables of the manuscript, green ticks indicate approaches involving inter-modal similarity comparisons, i.e. similarity comparisons between features of two different modalities (such as image-text, OTI-image, and OVI-text). Red crosses, on the other hand, refer to methods that employ intra-modal comparisons (such as image-image, text-text, OTI-OTI, and OVI-OVI).
> >
> > In Table 2 (right) the inter-modal baseline (corresponding to white rows) considers inter-modal similarity comparisons between the input image and the textual class prompts. When applying OTI to the input image (corresponding to blue rows) we compare the OTI-inverted features with the textual class prompts. Since OTI maps the visual features of the input image to the textual embedding space, such an approach involves intra-modal similarity comparisons. Results show that as expected, transforming an inter-modal task into an intra-modal one *decreases* the performance over the inter-modal baselines due to the intra-modal misalignment.
> >
> > > Q1: Does combining the native image representation with its inverted (cross-modal) image representation provide any performance benefits?
> >
> > This is a very interesting suggestion. We conducted an image retrieval experiment to assess whether combining native image features with the corresponding OTI-inverted features improves the performance. We define the native image features as $\psi_I = f_{\theta}(I)$ and the OTI-inverted features as $\psi_T = g_{\phi}(Y_{v^*})$. To query the gallery, we use a weighted combination of these two representations: $\psi_{IT} = \alpha * \psi_T + (1 - \alpha) * \psi_I$.
> >
> >
> > The table below reports the results for varying values of the parameter $\alpha$ for image-to-image retrieval:
> >
> > |Method|CUB|SOP|ROxf.|RParis|Cars|AVG|
> > |--|:--:|:--:|:--:|:--:|:--:|:--:|
> > | Intra-modal ($\alpha=0$) |22.9|34.4|42.6|67.9|24.6|38.5|
> > | OTI ($\alpha=0.25$) |24.0|35.6|44.9|70.1|25.9|40.1|
> > | OTI ($\alpha=0.50$) |24.6|**36.1**|**46.7**|71.0|27.0|41.1|
> > | OTI ($\alpha=0.75$) |**24.8**|35.9|46.3|**71.1**|27.7|**41.2**|
> > | **OTI (ours)** ($\alpha=1$)|24.6|35.1|43.0|70.3|**28.0**|40.2|
> >
> >
> > These results are very interesting (and surprising). Combining $\psi_I$ and $\psi_T$ consistently improves performance across all image retrieval datasets compared to the intra-modal baseline. Notably, when $\alpha$ is set to 0.5 or 0.75, the combination even outperforms OTI features on 4 out of 5 image retrieval datasets. The best average performance is achieved with $\alpha = 0.75$. This insight certainly warrants further exploration and investigation, for example on how to select $\alpha$ at test time, and we will add elements of this analysis to the final revision of the manuscript.
> >
> > **(Continued in next message)**

---

> > > ### Author Response · Authors · 2024-11-20
> > > **Response to Reviewer BNBn (part 3).**
> > >
> > > > Q2:  For text-to-text retrieval, have you tried the method on a purely textual task with less than 77 tokens context? Existing text retrieval datasets could be summarized to 77 tokens using an LLM. My assumption is that this approach would only be effective when the text can be represented visually, which might explain its success with text derived from image-captioning datasets.
> > >
> > > We followed the Reviewer's suggestion and tested our approach on purely textual text-to-text retrieval datasets. Specifically, we selected seven datasets from the [NanoBEIR](https://huggingface.co/collections/zeta-alpha-ai/nanobeir-66e1a0af21dfd93e620cd9f6) benchmark. These datasets cover diverse domains, ranging from scientific documents (SciDOCS) to texts related to climate change (ClimateFEVER). For this selection we discarded Question-Answering (QA) datasets and those with queries whose average length exceeds 77 tokens. To further expand the analysis, we also experimented with the IMDB Reviews [5] and 20 Newsgroups [6] datasets.
> > >
> > > All the datasets comprise query texts (which are inverted using OVI) that cannot be easily represented visually. Some query text examples are: “Learning Actionable Representations with Goal-Conditioned Policies” (SciDocs), “Atheism, philosophy, and the absence of belief in deities” (20 Newsgroup), and “The carbon footprint on wind energy is significant” (ClimateFEVER).
> > >
> > > Following the Reviewer’s suggestion, we used an LLM (the [meta-llama/Llama-3.2-1B-Instruct](https://huggingface.co/meta-llama/Llama-3.2-1B-Instruct) model) to summarize the *gallery* texts to 77 or fewer tokens. The following table reports the retrieval results in terms of mAP using the CLIP ViT-B/32 model:
> > >
> > > | Method | Inter modal | IMDB | 20Newsgroups | Climate | DBPedia | FEVER | NFCorpus | NQ | SciDocs | SciFact | AVG |
> > > |----------|:----------:|:------:|:----------:|:------:|:------:|:------:|:------:|----------|----------|----------|----------|
> > > | Intra-modal baseline | ❌ | 52.2 | 19.2 | 11.2 | 30.3 | 58.4 | 8.9 | 23.3 | 13.5 | 26.3 | 27.0 |
> > > | **OVI** | ✅ | **52.3** | **33.1** | **15.3** | **39.1** | **70.5** | **12.2** | **33.6** | **16.8** | **33.2** | **34.0** |
> > >
> > > These results show that the performance improvement from using modality inversion (OVI) is consistent across these datasets as well. This suggests that OVI is effective even in cases where the text cannot be easily represented visually.
> > >
> > > > Q4: Do you have any intuitions on why the image-image retrieval score for EuroSat dataset goes down with modality inversion?
> > >
> > >
> > > We hypothesize that the EuroSAT dataset, consisting of satellite images, represents a significant domain shift from the images CLIP was trained on. These images are challenging to describe accurately with text, and it's difficult to highlight subtle differences between them through textual descriptions. As a result, the modality inversion via OTI does not provide the same benefits as with other datasets.
> > >
> > > [1] Li et al. "DeCap: Decoding CLIP Latents for Zero-Shot Captioning via Text-Only Training." arXiv 2023.
> > >
> > > [2] Yu et al. "CoCa: Contrastive Captioners are Image-Text Foundation Models." arXiv 2022.
> > >
> > > [3] Liu et al. "Visual Instruction Tuning." NeurIPS 2024.
> > >
> > > [4] Patel et al. "ECLIPSE: A Resource-Efficient Text-to-Image Prior for Image Generations." CVPR 2024.
> > >
> > > [5] Maas et al. "Learning Word Vectors for Sentiment Analysis." ACV 2011.
> > >
> > > [6] Lang "NewsWeeder: Learning to Filter Netnews." Machine learning 1995

---

> ### Comment · Reviewer_BNBn · 2024-11-25
>
> Thank you for the clarifying experiments. All my questions have been sufficiently addressed. I think this paper addresses an important problem (poor intra-model performance of CLIP), provides a novel analysis of why this happens (due to the inter-modal CLIP loss and the modality gap), and a solution (OVI, OTI features). I also believe that demonstrating how intra-modal losses help address the modality gap problem is important for advancing the field.
>
> I maintain my rating and recommend accepting the paper for publication.

---

> > ### Author Response · Authors · 2024-11-25
> > **Thank you.**
> >
> > We thank the Reviewer again for their feedback and for recommending acceptance of our submission. The suggested additional experiments and clarifications have helped us improve our manuscript.

---

### Official Review · Reviewer_ikx3 · 2024-11-02

**Soundness:** 3
**Presentation:** 3
**Contribution:** 3
**Rating:** 8
**Confidence:** 5

**Summary:**

The authors point out in this paper that the common practice of using intra-modal similarities from pre-trained CLIP encoders is inherently suboptimal for intra-modal tasks. To address this, they adapt optimization-based modality inversion techniques (OTI and OVI) and leverage them to transform native modality inputs into intermodal representations for similarity measures. Experiments across three settings—image-to-image retrieval, text-to-text retrieval, and zero-shot image classification—on 15 datasets and with 5 different pre-trained models validate the authors’ hypothesis, demonstrating the effectiveness of optimization-based modality inversion.

**Strengths:**

- The idea is good and interesting. Using inter-modal CLIP representations for intra-modal similarity measures makes sense. If this has not been done before, it is a novel idea.

- The paper is well-motivated, with comprehensive experiments, promising results, and clear writing.

- The authors validate their hypothesis and demonstrate their method's effectiveness through extensive experiments involving over 15 datasets, 3 different types of CLIP models, and 5 different pre-trained models.

**Weaknesses:**

- The method is quite simple. OVI is similar to OTI. There is nothing new about the inversion. But applying the inversion in this particular context can be appreciated. The authors might want to tune down the claim on the methodology contribution.

- It is a surprise to see the proposed method also helps with image classification tasks. Why? Image classification is supposed to be an inter-modal task. In the open-vocabulary setting, it is often done by comparing CLIP image features with a set of CLIP text features on the class vocabulary, which is already inter-modal.

- Missing baseline: For the image-to-image retrieval task, the authors convert the query image to a corresponding textual feature using OTI, then calculate semantic similarity between this feature and the other image features (obtained from the CLIP image encoder) to get the final results. An important baseline is missing, i.e., converting all the images into OTI features and then measuring the similarity among OTI features. This can demonstrate your proposed inter-modal CLIP representations are better for intra-modal tasks.

**Questions:**

- Please answer those issues pointed out in the weakness.
- In Table 1, the use of OTI features outperforms native image features on all datasets except for the EuroSAT dataset. What distinguishes the EuroSAT dataset from the others? Why does it show a performance decrease with OTI features?
- The authors use the template sentence “a photo of” concatenated with the pseudo-word tokens in Section 4.1. Have you tried other templates?
- It would be better to provide more training details (e.g., training time, memory usage, and training data).

---

> ### Author Response · Authors · 2024-11-20
> **Response to Reviewer ikx3 (part 1).**
>
> We thank the Reviewer for their thoughtful review and for their recognition of the comprehensiveness of our experimental evaluation and the strong motivations for using inter-modal CLIP representations for intra-modal similarity comparisons. Below we respond to each specific point raised by the Reviewer.
>
> > W1: The method is quite simple. OVI is similar to OTI. There is nothing new about the inversion. But applying the inversion in this particular context can be appreciated. The authors might want to tune down the claim on the methodology contribution.
>
> Our main claim is that ours is the first work to our knowledge demonstrating the negative effects of intra-modal misalignment in pre-trained CLIP encoders when applied to intra-modal problems like image-to-image and text-to-text retrieval, and that this misalignment can be mitigated by deriving inter-modal features by exploiting the encoder of the complementary modality. This contribution is supported by a comprehensive study of intra-modal misalignment in CLIP and the extensive experiments showing that using inter-modal representations derived via modality inversion can significantly improve the performance over intra-modal baselines.
>
> We agree that modality inversion is not the main novel contribution of our work, it is rather a tool we use in our analysis to demonstrate that inter-modal features can be derived for intra-modal tasks. OTI, as introduced in [1], was used to combine inputs from both modalities (i.e. an input text and an input image image). In our work we use it for the very different goal of mapping a single modality to its complementary one. That is, as a mapping technique from visual to textual representations (OTI), or from textual to visual representations (OVI). The novelty in our application of modality inversion lies in the way we apply it to cast intra-modal tasks into inter-modal ones.
>
> > W2: It is a surprise to see the proposed method also helps with image classification tasks. Why? Image classification is supposed to be an inter-modal task. In the open-vocabulary setting, it is often done by comparing CLIP image features with a set of CLIP text features on the class vocabulary, which is already inter-modal.
>
> In fact, modality inversion **does not help** with image classification tasks. In Table 2 (right) the rows with the *green check marks* correspond to the zero-shot, open-vocabulary setting using the CLIP image and text encoders. The rows highlighted in blue and indicated with *red Xs* instead correspond to using OTI-inverted features for query images and indeed perform **worse** than zero-shot CLIP given that -- as the Reviewer correctly observes -- this is an inherently inter-modal task.
>
> We performed this experiment to show that the performance improvement on intra-modal tasks stems from leveraging inter-modal alignment and *not* from modality inversion itself. Applying modality inversion to such an inherently inter-modal task as zero-shot image classification transforms it into an intra-modal one and thus we expect a performance *decrease* due to intra-modal misalignment. Indeed, OTI maps the visual features of the input image to the textual embedding space, and thus comparing them with the textual features of the prompts makes the task intra-modal. Tab. 2 (right) confirms our hypothesis. To further confirm this claim, in Appendix F we show that the same considerations apply also to the other inherently inter-modal task of image-text retrieval.
>
> We will revise the manuscript to improve the clarity and discussion of these intra-modal results.
>
> > W3: Missing baseline: For the image-to-image retrieval task, the authors convert the query image to a corresponding textual feature using OTI, then calculate semantic similarity between this feature and the other image features (obtained from the CLIP image encoder) to get the final results. An important baseline is missing, i.e., converting all the images into OTI features and then measuring the similarity among OTI features. This can demonstrate your proposed inter-modal CLIP representations are better for intra-modal tasks.
>
> To address this point, we conducted the suggested experiment in which we inverted all images using OTI and measured the similarity among these OTI features for image-to-image retrieval. The results are summarized in the table below:
>
> | Method | Inter modal | CUB | SOP | ROxf. | RParis | Cars | AVG |
> |--|:--:|:--:|:--:|:--:|:--:|:--:|:--:|
> | Intra-modal baseline | ❌ | 22.9 | 34.4 | 42.6 | 67.9 | 24.6 |38.5|
> | Intra-OTI |❌|21.3|31.9|42.3|68.2|24.9|37.7|
> | **OTI (ours)** |✅|**24.6**|**35.1**|**43.0**|**70.3**|**28.0**|**40.2**|
>
> As shown, computing similarities among OTI-inverted features (Intra-OTI) results in performance that is slightly worse than the intra-modal baseline. This indicates that intra-modal misalignment persists even when both query and gallery images are represented in the text embedding space via OTI.
>
> **(Continued in next message)**

---

> > ### Author Response · Authors · 2024-11-20
> > **Response to Reviewer ikx3 (part 2).**
> >
> > As discussed in our previous response to **W1** and **W2**, our primary contribution is demonstrating that leveraging CLIP's inter-modal alignment can enhance performance on intra-modal tasks. By mapping the query image to the text embedding space using OTI and comparing it with the original image features of the gallery, we are effectively performing inter-modal similarity comparisons between different modalities (i.e. text and image, respectively). This approach takes advantage of the strong inter-modal alignment inherent in CLIP models. On the contrary, converting all images into the same modality and comparing them, as in the Intra-OTI baseline, decreases the performance by keeping the task intra-modal.
> >
> > These results confirm that inter-modal similarity comparisons outperform intra-modal ones, and we will clarify this point in the revised manuscript to ensure that the distinction between using inter-modal representations (comparing features from different modalities) and intra-modal representations (comparing features of the same modality) is clear.
> >
> > > Q2: In Table 1, the use of OTI features outperforms native image features on all datasets except for the EuroSAT dataset. What distinguishes the EuroSAT dataset from the others? Why does it show a performance decrease with OTI features?
> >
> > We appreciate this insightful question. We hypothesize that the EuroSAT dataset, consisting of satellite images, represents a significant domain shift from the images CLIP was trained on. These images are challenging to describe accurately with text, and it is difficult to highlight subtle differences between them through textual descriptions. As a result, the modality inversion via OTI does not provide the same benefits as with other datasets.
> >
> > > Q3: The authors use the template sentence “a photo of” concatenated with the pseudo-word tokens in Section 4.1. Have you tried other templates?
> >
> > In our preliminary experiments we tested various template sentences and observed minimal performance differences between them. To quantitatively and rigorously validate this observation, we conducted an experiment comparing the performance of different prompts on image retrieval datasets using the CLIP ViT-B/32 model. Specifically, we tested the following prompts:
> > - “an image of $v^*$”,
> > - “we see a $v^*$ in this photo”, and
> > - “$v^*$” (the *empty prompt*),
> >
> > where $v^*$ is the learnable pseudo-word token.
> >
> > The results expressed in terms of mAP, summarized in the table below, confirm our preliminary findings, showing robustness and minimal differences in performance across the different prompts.
> >
> > | Method | Inter modal | CUB | SOP | ROxf. | RParis | Cars | AVG |
> > |-----------------------------------|:----------:|:------:|:------:|:------:|:------:|:------:|:------:|
> > | Intra-modal baseline | ❌ | 22.9 | 34.4 | 42.6 | 67.9 | 24.6 | 38.5 |
> > | OTI ("$v^{*}$") | ✅ | 24.0 | 34.6 | **43.7** | 69.6 | 28.2 | 40.0 |
> > | OTI ("we see $v^{*}$ in this photo") | ✅ | 24.5 | 34.7 | 43.0 | 69.7 | **28.3** | 40.0 |
> > | OTI ("an image of $v^{*}$") | ✅ | 24.0 | 34.8 | 43.1 | **70.7** | *28.3* | **40.2** |
> > | **OTI (ours)** ("a photo of $v^{*}$") | ✅ | **24.6** | **35.1** | 43.0 | 70.3 | 28.0 | **40.2** |
> >
> >
> > > Q4: It would be better to provide more training details (e.g., training time, memory usage, and training data).
> >
> > In Appendix A we have already provided a detailed list of implementation details, including the training time for both OTI and OVI using a single A100 GPU and the CLIP ViT-B/32 model.
> >
> > Regarding memory usage, it scales linearly with the batch size. Specifically, when using the CLIP ViT-B/32 model, we observe that OTI requires approximately 1,878 MiB plus 18.6 MiB per sample in the batch. For example, with a batch size of 128, the memory consumption is around 4,260 MiB. For OVI, the memory usage is approximately 2,218 MiB plus 16.2 MiB per sample, resulting in about 4,290 MiB with the same batch size. We will include this analysis of memory usage in Appendix A of the revised manuscript.
> >
> > Since the modality inversion techniques we employ operate at the single-feature level, they independently map each feature to the complementary modality without requiring external training data. Details about the benchmark datasets used in our experiments are available in Appendix E.
> >
> > [1] Baldrati et al. "Zero-Shot Composed Image Retrieval with Textual Inversion." ICCV 2023.

---

> > > ### Comment · Reviewer_ikx3 · 2024-11-28
> > > **Reviewer response to the rebuttal**
> > >
> > > The rebuttal has well-addressed my concerns. I am happy to recommend an acceptance.

---

> > > > ### Author Response · Authors · 2024-11-29
> > > > **Thank you.**
> > > >
> > > > We thank the Reviewer again for their helpful feedback. We are pleased that the rebuttal has addressed their concerns and we appreciate their positive evaluation. The suggested additional clarifications and experiments have certainly helped improve the quality of our manuscript.

---

### Official Review · Reviewer_U71q · 2024-11-03

**Soundness:** 2
**Presentation:** 3
**Contribution:** 2
**Rating:** 6
**Confidence:** 4

**Summary:**

The paper points out the problem of  intra-modal misalignment in CLIP, which limits its effectiveness for intra-modal tasks such as image-to-image retrieval. To address this issue, the paper proposes to transform intra-modal tasks to inter-modal ones via Optimization-based Textual Inversion (OTI) and Optimization-based Visual Inversion (OVI), which map representations to the complementary modality. The experiments demonstrate its effectiveness on fifteen datasets.

**Strengths:**

1. The paper tackles an intriguing problem, since CLIP encoders are widely used in various tasks and serves as components in other models like LVLM. Addressing the intra-modal misalignment issue could enhance CLIP's performance in image and text understanding.
2. The presentation quality is high. The paper is well-written, easy to follow, and features clear and informative figures.

**Weaknesses:**

1. The analysis of the intra-modal misalignment problem could benefit from more quantitative insights. In Figure 1, the authors illustrate that intra-modal similarity scores can be higher for the same class than for different ones. Providing statistics to demonstrate the significance and frequency of this issue would strengthen the argument.
2. The experiments lack comparative analysis. While the related works section mentions existing works that explore intra-modal misalignment, it would be beneficial to compare these methods in the experimental study to validate the paper's contributions.

**Questions:**

It is interesting to know if it is possible to have a more systematic approach for selecting the number of tokens and optimization steps used in OTI and OVI, as discussed in Figure 2.

---

> ### Author Response · Authors · 2024-11-20
> **Response to Reviewer U71q (part 1).**
>
> We thank the Reviewer for their thoughtful review, for their recognition that we address an intriguing problem, and for appreciating the presentation quality and clarity of our contribution. Below we respond to specific points raised.
>
> > W1: The analysis of the intra-modal misalignment problem could benefit from more quantitative insights. In Figure 1, the authors illustrate that intra-modal similarity scores can be higher for the same class than for different ones. Providing statistics to demonstrate the significance and frequency of this issue would strengthen the argument.
>
> To provide concrete evidence of intra-modal misalignment and demonstrate that intra-modal similarity scores can sometimes be *lower* for the same class than for different ones, we conducted a simple experiment using CLIP ViT-B/32 and the “dogs vs cats” dataset [1] consisting of 25K images equally distributed between dog and cat classes.
>
> We first filtered out dog images that are not inter-modally aligned, i.e. dog images that are more similar to the prompt “a photo of a cat” than to the prompt “a photo of a dog”. Then we filtered out cat images analogously. This filtering ensures a focus on samples for which inter-modal alignment is correct. On this filtered dataset, we computed the similarities between dog images (queries) and the entire dataset. Since we exclude inter-modally misaligned samples, if inter-modal alignment implies intra-modal alignment, retrieval should be perfect -- that is, all dog images should rank *higher* than cat images for each query. However, our results show that this is not the case. Specifically, we observe a mean Average Precision (mAP) of 81.4% and an average R-Precision of 71.5%, where R-Precision is the precision at rank R, with R being the total number of relevant items for a given query. This means that in almost 30% of cases, cats are ranked *higher* than other dogs for a given dog query. A similar result was observed for cats.
>
> We are revising the manuscript in order to incorporate this analysis -- along with a histogram visualization similar to that in Fig. 2 \(c\) -- into the main paper as we feel it indeed helps support our claim and intuitive motivation illustrated in Fig. 1. We thank the Reviewer for stimulating reflection on this.
>
> > W2: The experiments lack comparative analysis. While the related works section mentions existing works that explore intra-modal misalignment, it would be beneficial to compare these methods in the experimental study to validate the paper's contributions.
>
> Existing works that explore the problem of intra-modal misalignment [2, 3] address only the zero- and few-shot image classification tasks. Such methods require the knowledge of class names to build a set of support images [2] or to generate auxiliary texts [3]. For this reason, these approaches are not applicable to image-to-image or text-to-text retrieval.
>
> To address the Reviewer’s concern about the lack of comparative analysis we performed experiments using additional inter-modal baselines.
>
> #### Inter-modal Representations via Adapters
>
> We trained adapters to map features from their native modality to the complimentary ones (as suggested by Reviewer pj7m). To train each adapter we used the LLaVA-CC3M dataset [4], which comprises 595K image-text pairs. Adapters are trained using a cosine loss to minimize the distance between the adapter output and the corresponding complementary features. Additionally, following Patel et al. [5], we incorporated a CLIP-based contrastive loss during training. We trained two separate adapters: one for mapping image features to text features (aligned with the goal of OTI) and another for mapping text features to image features (aligned with the goal of OVI).
>
> First for image-to-image retrieval:
>
> | Method | Inter modal | CUB | SOP | ROxf. | RParis | Cars | AVG |
> |--|:--:|:--:|:--:|:---:|:--:|:--:|:--:|
> | Intra-modal baseline | ❌ | 22.9 | 34.4 | 42.6 | 67.9 | 24.6 | 38.5 |
> | Adapter | ✅ | 23.7 | 35.0 | *44.3* | 69.5 | 25.5 | 39.6 |
> | *OTI (ours)* | ✅ | *24.6* | *35.1* | 43.0 | *70.3* | *28.0* | *40.2* |
>
> And for text-to-text retrieval:
>
> | Method | Inter modal | Flickr30k | COCO | nocaps | AVG |
> |--|:--:|:--:|:--:|:--:|:--:|
> | Intra-modal baseline | ❌ | 51.7 | 26.2 | 35.1 | 37.7 |
> | Adapter | ✅ | 51.9 | **28.3** | 37.8 | 39.3 |
> | **OVI (ours)** | ✅ | **54.8** | **28.3** | **38.8** | **40.6** |
>
> The adapter approach improves over the intra-modal baseline for both text and image retrieval tasks, which aligns with our hypothesis that leveraging inter-modal representations for intra-modal tasks enhances performance due to CLIP’s inherent inter-modal alignment. However, on average OTI and OVI outperform the adapter-based approach. This finding is noteworthy given that OTI and OVI **do not require an additional training dataset**. Instead, they map individual features directly to the complementary modality without relying on external resources.
>
> **(Continued in next message)**

---

> > ### Author Response · Authors · 2024-11-20
> > **Response to Reviewer U71q (part 2).**
> >
> > #### Inter-modal Features via Captioning
> >
> > We also investigated the performance of image-to-image retrieval using a captioning model to convert the query image into descriptive text (as suggested by Reviewer BNBn). Given a query image, we generated a caption using a pre-trained captioning model, extracted text features from the generated caption using the CLIP text encoder, and used these features to perform retrieval.
> >
> > We experimented with three captioning models:
> > 1. DeCap [6], which directly generates captions from CLIP image features, making it the most comparable approach since OTI also relies only on CLIP image features;
> > 2. CoCa ViT-B/32 (LAION) [7], trained on the Laion2B dataset; and
> > 3. CoCa ViT-B/32 (MSCOCO) [7], pre-trained on Laion2B and fine-tuned on MSCOCO.
> >
> > The table below summarizes the image retrieval results in terms of mAP:
> >
> > | Method | Inter modal | CUB | SOP | ROxf. | RParis | Cars | AVG |
> > |--|:--:|:--:|:--:|:--:|:--:|:--:|:--:|
> > | Intra-modal baseline | ❌ | 22.9 | 34.4 | 42.6 | 67.9 | 24.6 | 38.5 |
> > | DeCap | ✅ | 4.4 | 2.0 | 0.1 | 1.2 | 2.5 | 2.0 |
> > | CoCa (MSCOCO) | ✅ | 3.5 | 0.8 | 0.0 | 0.7 | 1.8 | 1.4 |
> > | CoCa (LAION) | ✅ | 17.6 | 3.9 | 8.4 | 28.2 | 23.6 | 16.3 |
> > | **OTI (ours)** | ✅ | **24.6** | **35.1** | **43.0** | **70.3** | **28.0** | **40.2** |
> >
> > In all cases captioning-based retrieval falls short of the performance of the intra-modal baseline, despite their leveraging of CLIP’s image-text alignment. Furthermore, the effectiveness of captions varies with the dataset domain: captioners struggle to produce discriminative captions for datasets featuring buildings (ROxford and RParis) while achieving better results in domains like cars (Cars). Inter-modal features derived via modality inversion (OTI), on the other hand, improve image retrieval performance on all datasets.
> >
> > To understand the variability in performance of different captioning models, we report captions generated by the three models for a randomly chosen image (“all_souls_000026”) from the ROxford dataset depicting the All Souls College:
> >
> > - DeCap: “a large building with a clock tower on the front.”;
> > - CoCa (MSCOCO): “an old building with two towers has a clock on it.”; and
> > - CoCa (LAION): “all souls college, oxford, united kingdom.”
> >
> > This example shows that the first two models fail to generate sufficiently discriminative captions, while CoCa (LAION) produces a more precise description, correlating with its higher performance among the captioning models.
> >
> > We hypothesize that a more advanced captioner, such as a large multimodal language model (e.g. ChatGPT-4V), could generate more precise descriptions and potentially improve retrieval performance by better leveraging cross-modal alignment. However, comparing such an approach with OTI, which does not rely on any external data, would not be fair.
> >
> > We will add these new inter-modal baseline comparisons to the revised manuscript.
> >
> > > Q1: It is interesting to know if it is possible to have a more systematic approach for selecting the number of tokens and optimization steps used in OTI and OVI, as discussed in Figure 2.
> >
> > We agree that a more systematic approach for selecting the number of tokens and optimization steps in OTI and OVI would be beneficial. In our experiments, we observed that the performance of OTI improves with the number of optimization steps up to around 100 iterations, after which it stabilizes or may slightly decrease (as shown in Figure 2(b)). This observation led us to choose 150 optimization steps for all OTI results reported. A similar pattern was noted for OVI. We mentioned this in our Limitations section, but we will strengthen and clarify this discussion in the revised manuscript.
> >
> > Regarding the number of tokens, for OTI we found that using a single pseudo-word token is sufficient across different tasks, and the performance becomes robust after a certain number of optimization steps, consistently improving over the intra-modal baseline. For OVI, the required number of pseudo-patches depends on the expressiveness of the backbone architecture (i.e., the image encoder), as detailed in Appendix C. We will provide guidelines and include these findings in the revised manuscript to assist practitioners in selecting appropriate hyperparameters.
> >
> > [1] Elson et al. "Asirra: A CAPTCHA that Exploits Interest-Aligned Manual Image Categorization." CCS 2007.
> >
> > [2] Udandarao et al. "SuS-X: Training-Free Name-Only Transfer of Vision-Language Models." ICCV 2023.
> >
> > [3] Yi et al. "Leveraging Cross-Modal Neighbor Representation for Improved CLIP Classification." CVPR 2024.
> >
> > [4] Liu et al. "Visual Instruction Tuning." NeurIPS 2024.
> >
> > [5] Patel et al. "ECLIPSE: A Resource-Efficient Text-to-Image Prior for Image Generations." CVPR 2024.
> >
> > [6] Li et al. "DeCap: Decoding CLIP Latents for Zero-Shot Captioning via Text-Only Training." arXiv 2023.
> >
> > [7] Yu et al. "CoCa: Contrastive Captioners are Image-Text Foundation Models." arXiv 2022.

---

> > > ### Comment · Reviewer_U71q · 2024-11-26
> > >
> > > Thank the authors for providing a detailed response, most of my concerns have been addressed. I suggest including more examples in the quantitative analysis of intra-modal misalignment (revised Fig. 2) in the future version, such as more classes, to provide more comprehensive support for their claim and motivation.
> > >
> > > I will raise my rating in favor of accepting this paper.

---

> > > > ### Author Response · Authors · 2024-11-26
> > > > **Thank you.**
> > > >
> > > > We thank the Reviewer again for their thoughtful feedback -- and especially for suggesting to include quantitative evidence of the misalignment phenomenon -- and for increasing their score in favor of accepting our submission. We agree with the suggestion to expand this quantitative analysis and we will study how to best accomplish this in the final version of our work.

---

### Official Review · Reviewer_pj7m · 2024-11-04

**Soundness:** 2
**Presentation:** 3
**Contribution:** 2
**Rating:** 3
**Confidence:** 5

**Summary:**

This paper investigates the suboptimal performance of using pre-trained multi-modal Vision Language Models (VLMs) like CLIP for intra-modal tasks. It highlights the modality gap, a disparity between text and image feature spaces due to contrastive pre-training, which leads to poor performance in tasks like image-to-image retrieval when using individual text or image encoders. The authors propose a modality inversion technique to transform native modality inputs into inter-modal representations, leveraging CLIP's inter-modal alignment. Extensive experiments demonstrate significant performance improvements over intra-modal baselines in image retrieval, text retrieval, and zero-shot image classification tasks.

**Strengths:**

- The paper introduces a novel modality inversion method that significantly outperforms standard intra-modal approaches by exploiting inter-modal alignments of CLIP models, providing a new direction for enhancing VLMs' utility in single-modal tasks.

-  With experiments spanning multiple datasets and tasks, the paper offers robust evidence of the proposed method's efficacy, showcasing its broad applicability in the field of multi-modal learning.

**Weaknesses:**

- The contribution is incremental, as the OTI has already been introduced by existing methods.
- In zero-shot classification, current methods utilize both image and text encoders concurrently. Therefore, only utilizing the image encoder compromises fairness in the experimental comparisons for this task.
- Several relevant comparative methods are omitted. In particular, the proposed method achieves similar effects to prompt learning and adapter learning, which should be discussed and included in further comparisons.

**Questions:**

Please see the Weaknesses.

---

> ### Author Response · Authors · 2024-11-20
> **Response to Reviewer pj7m (pt. 1)**
>
> We thank the Reviewer for their thoughtful review and for recognizing the novelty of our contribution, that our analyses provide a new direction for enhancing VLM performance on single-modal tasks, and the robustness of our experimental evaluation. Below we respond to each specific point raised by the Reviewer.
>
> > W1: The contribution is incremental, as the OTI has already been introduced by existing methods.
>
> We respectfully disagree with the assertion that our contribution is incremental. Ours is the first work to our knowledge demonstrating the negative effects of intra-modal misalignment in pre-trained CLIP encoders when applied to intra-modal problems like image and text retrieval. This contribution is supported by a comprehensive study of intra-modal misalignment in CLIP and the extensive experiments showing that using inter-modal representations derived via modality inversion can significantly improve the performance over intra-modal baselines.
>
> Optimization-based Textual Inversion (OTI), as introduced in [1], was used to combine inputs from **both** modalities (i.e. an input text and an input image). In our work we use it in a different setting and for the very different goal of mapping a single modality to its complementary one. That is, as a mapping technique from visual to textual representations, as explained in L246-L255. The novelty in our application of modality inversion (both OTI and the proposed OVI) lies in the way we apply it to cast intra-modal tasks into inter-modal ones.
>
> > W2: In zero-shot classification, current methods utilize both image and text encoders concurrently. Therefore, only utilizing the image encoder compromises fairness in the experimental comparisons for this task.
>
> In zero-shot image classification, the inter-modal baseline (i.e. the first evaluation setting described in L409-L411 whose results are reported in the white rows of the right section of Tab. 2) utilizes **both** the image and the text encoder by comparing the features of the input image with those of the textual prompts.  When applying OTI to the input image (i.e. the second evaluation setting described in L409-L411 whose results are reported in the blue rows of the right section of Tab. 2) we compare the OTI-inverted features with the textual prompts. Since OTI employs both encoders, the fairness of the experimental setting is ensured.
>
> We performed this experiment to show that the performance improvement that we achieve on intra-modal tasks stems from leveraging inter-modal alignment and *not* from the modality inversion process itself, as explained in L83-L85. Applying modality inversion to such an inherently inter-modal task as zero-shot image classification transforms it into an intra-modal one and thus we expect a performance *decrease* due to intra-modal misalignment (see L314-L316 and L405-L407). Indeed, OTI maps the visual features of the input image to the textual embedding space, and thus comparing them with the textual features of the prompts makes the task intra-modal. Tab. 2 (right) confirms our hypothesis, as detailed in L412-L423. To further confirm this claim, in Appendix F we show that the same considerations apply also to the other inherently inter-modal task of image-text retrieval.

---

> > ### Author Response · Authors · 2024-11-20
> > **Response to Reviewer pj7m (pt. 2)**
> >
> > > W3: Several relevant comparative methods are omitted. In particular, the proposed method achieves similar effects to prompt learning and adapter learning, which should be discussed and included in further comparisons.
> >
> > Both prompt learning and adapter learning require additional parameters and -- more importantly -- an additional dataset of image-text pairs for training. Our proposed modality inversion approach works at the single-instance level and requires no additional training data.
> >
> > However, to broaden our comparative analysis we conducted an additional experiment in which we trained single-layer adapters to map features from the native modality to the complementary modality. To train each adapter we used the [LLaVA-CC3M](https://huggingface.co/datasets/liuhaotian/LLaVA-CC3M-Pretrain-595K) dataset [2], which comprises 595K image-text pairs. This dataset is derived by filtering the CC3M dataset [3] to achieve a more balanced distribution of concept coverage. Adapters are trained using a cosine loss to minimize the distance between the adapter output and the corresponding complementary features. Additionally, following Patel et al. [4], we incorporated a CLIP-based contrastive loss during training. We trained two separate adapters: one for mapping image features to text features (aligned with the goal of OTI) and another for mapping text features to image features (aligned with the goal of OVI).
> >
> > The results, evaluated using the CLIP ViT-B/32 model on both image and text retrieval datasets, are summarized in the tables below. First, for image-to-image retrieval:
> >
> > | Method | Inter modal | CUB | SOP | ROxf. | RParis | Cars | AVG |
> > |---------------|:----------:|:------:|:------:|:------:|:------:|:------:|:------:|
> > | Intra-modal baseline | ❌ | 22.9 | 34.4 | 42.6 | 67.9 | 24.6 | 38.5 |
> > | Adapter | ✅ | 23.7 | 35.0 | **44.3** | 69.5 | 25.5 | 39.6 |
> > | **OTI (ours)** | ✅ | **24.6** | **35.1** | 43.0 | **70.3** | **28.0** | **40.2** |
> >
> > And for text-to-text retrieval:
> >
> > | Method | Inter modal | Flickr30k | COCO | nocaps | AVG |
> > |---------------|:----------:|:-------:|:------:|:------:|:------:|
> > | Intra-modal baseline | ❌ | 51.7 | 26.2 | 35.1 | 37.7 |
> > | Adapter | ✅ | 51.9 | **28.3** | 37.8 | 39.3 |
> > | **OVI (ours)** | ✅ | **54.8** | **28.3** | **38.8** | **40.6** |
> >
> > The adapter approach improves performance over the intra-modal baseline for both text and image retrieval tasks. This aligns with our hypothesis that leveraging inter-modal representations for intra-modal tasks enhances performance due to CLIP’s inherent inter-modal alignment. However, on average OTI and OVI outperform the adapter-based approach. This finding is particularly noteworthy given that OTI and OVI do not require a training dataset. Instead, they map individual features directly to the complementary modality without relying on external resources.
> >
> > We thank the Reviewer for this suggestion and will integrate these results into the revised manuscript.
> >
> > [1] Baldrati et al. "Zero-Shot Composed Image Retrieval with Textual Inversion." ICCV 2023.
> >
> > [2] Liu et al. "Visual Instruction Tuning." NeurIPS 2024.
> >
> > [3] Sharma et al. "Conceptual Captions: A Cleaned, Hypernymed, Image Alt-text Dataset For Automatic Image Captioning." ACL 2018.
> >
> > [4] Patel et al. "ECLIPSE: A Resource-Efficient Text-to-Image Prior for Image Generations." CVPR 2024.

---

> > > ### Comment · Reviewer_pj7m · 2024-11-25
> > >
> > > Thanks for your response. But my main concern is still not solved. According to your further explanation, the core contribution of this paper is the application of the existing OVI in different settings, which is sure an incremental work.

---

> > > > ### Author Response · Authors · 2024-11-25
> > > >
> > > > We thank the Reviewer for their engagement in the discussion and the additional feedback.
> > > >
> > > > We stress that, as articulated in our above response, our core contributions are not limited to the application of the *adapted* OTI and the *proposed* OVI in different settings. Our work identifies the problem of intra-modal misalignment in CLIP, providing valuable insights into its causes and ways to mitigate it. Our extensive experimental results support our claims. The links we draw between the misalignment phenomenon, the Modality Gap, and CLIP's inter-modal loss advance the understanding of VLMs. This is the fundamental contribution of our work, not simply a performance improvement on retrieval tasks.
> > > >
> > > > Moreover, we are perplexed by the reduction in rating from 5 to 3, despite addressing all three of the weaknesses pointed out in the original review. Are there any remaining concerns regarding W2 (the use of both encoders in relation to the fairness of our experimental evaluation) or W3 (the lack of comparative baselines)? We are happy to provide any additional clarifications and/or revisions if there are remaining doubts.

---

### Author Response · Authors · 2024-11-24
**Global Response and Revised Manuscript Upload**

We thank all reviewers for their insightful feedback aimed at improving the quality of our work. The reviewers agree that the paper is *well-written* (**U71q**, **ikx3**, **BNBn**) and *well-motivated* (**ikx3**), addresses an *intriguing problem* (**U71q**), and provides a *comprehensive and extensive* (**pj7m**, **ikx3**, **BNBn**) *empirical evaluation* spanning multiple datasets and tasks.

We have uploaded a revised version of the manuscript improving the clarity and including the changes suggested by the reviewers. We highlight in blue all the changes for clearer visualization. Please note that the paper lines we mentioned in the responses to the reviewers refer to the original paper and not the revised one. Below we provide a summary of the main additions to the revised manuscript.

### **Additional baselines (**pj7m**, **U71q**, **BNBn**)**
We expanded the "*Additional Experiments*" section (**Appendix F**) to include an adapter-based approach for image-to-image and text-to-text retrieval and a captioning-based method for image-to-image retrieval. Employing modality inversion techniques, as proposed in our work, achieves the best performance even after including these new baselines.

### **Quantitative insights on intra-modal misalignment (**pj7m**)**
We have added a new section to the main paper (**Sec. 2** of the revised manuscript) to provide quantitative insights on the impact of the intra-modal misalignment issue. We thank Reviewer **pj7m** for the suggestion, as we believe that this addition provides additional strength to our claims.

### **Additional Experiments (**BNBn**, **ikx3**)**
We included all the additional experiments suggested by the reviewers. Specifically, we added the following paragraphs to the "*Additional Experiments*" section (**Appendix F**):
- *Intra-OTI Similarity Comparisons* (**ikx3**): we applied OTI to all the images and measured the similarity among OTI-inverted features to perform image-to-image retrieval;
- *Impact of the OTI Template Sentence* (**ikx3**): we studied the impact on the performance of OTI template sentence on the image-to-image retrieval task;
- *Text-to-text Retrieval on Purely Textual Datasets* (**BNBn**): we evaluated the performance of OVI on purely textual datasets;
- *Combining Native and Inverted Features* (**BNBn**): we assessed wheter combining native and inverted features improves performance on image-to-image retrieval.

We again thank the reviewers for their thoughtful and insightful feedback. We are happy to address any further questions or provide additional clarifications.

---

### Author Response · Authors · 2024-12-02
**Final Global Response**

As the discussion period comes to a close, we would like to thank all reviewers again for their thoughtful and constructive feedback which has helped us improve the quality of our manuscript. We are still perplexed by Reviewer pj7m reducing their rating from 5 to 3, without further engaging in the discussion, despite our efforts to address all weaknesses mentioned in their original review. We remain open to any additional feedback that could help us improve our work.

---

### Meta-Review · Area_Chair_FSnU · 2024-12-17

**Metareview:**

The paper points out the problem of intra-modal misalignment in CLIP, which limits its effectiveness for intra-modal tasks such as image-to-image retrieval. To address this issue, the paper proposes to transform intra-modal tasks to inter-modal ones via Optimization-based Textual Inversion (OTI) and Optimization-based Visual Inversion (OVI), which map representations to the complementary modality. The experiments demonstrate its effectiveness on fifteen datasets. Major concerns of reviewers include lack of quantitative analysis of inter-modal misalignment, lack of baseline, and experimental reasonableness. The author solved the above problems during the rebuttal. So the final vote is acceptance.

**Additional Comments On Reviewer Discussion:**

The reviewer who voted rejection did not participate in the final discussion and did not ask new questions. Other reviewers approved the author's response and improved the score.

---

### Decision · Program_Chairs · 2025-01-22

Accept (Poster)